



# Measurement report: Emission factors and organic aerosol source apportionment of shipping emissions in the coastal city of Toulon, France

Quentin Gunti[1,2], Benjamin Chazeau[1], Brice Temime-Roussel[1], Irène Xueref-Remy[3], Alexandre Armengaud[2], Henri Wortham[1], and Barbara D'Anna[1]

[1]Aix Marseille Univ, CNRS, LCE, Marseille, France
[2]AtmoSud, Air Quality Regional Observatory in the South of France, Marseille, France
[3]Aix Marseille Univ., Avignon Université, CNRS, IRD, IMBE, Marseille, France

**Correspondence:** Quentin Gunti (quentin.gunti@univ-amu.fr) and Barbara D'Anna (barbara.danna@univ-amu.fr)

**Abstract.** Maritime transport has a significant impact on local air quality, especially in port areas. Ship emissions are recognized as major contributors to air pollution, comparable to road transport emissions. This study, conducted in 2021 in Toulon, a port city on the French Mediterranean coast, assessed emissions from shipping one year after the implementation of IMO2020 sulfur regulations. Emission factors (EFs) for pollutants such as $SO_2$, $NO_X$, CO, NO, $CH_4$ and PM as BC, organics (Org), $SO_4^{2-}$,

$NO_3^-$, $NH_4^+$ and PAHs were measured, as well as the particle number concentration (PN). IMO2020 regulation induced a significant reduction in sulfur-related emissions while other pollutants like soot, organics and PAHs remained at pre-regulation levels. Positive Matrix Factorization (PMF) of High-Resolution Time-of-Flight Aerosol Mass Spectrometer measurements of non-refractory $PM_1$ organic aerosol (OA) was used to investigate the shipping contribution to local air quality. PMF could separate road and marine transport emissions, revealing a shipping contribution to the total OA fraction of 11.2 %. Eight fac-

tors were resolved: three shipping-associated OA, a Hydrocarbon-like OA (HOA), a Cooking-like OA (COA), an Oxidized Hydrocarbon-like OA (OxHOA), a Less Oxidized OA (LOOA), and a More Oxidized OA (MOOA). Shipping and HOA factors were the major contributor to ultrafine particles and they represented the biggest emitter of alkylated PAHs (APAHs) (51.9 %). These findings underscore the importance of distinguishing shipping emissions in port areas and advanced source apportionment methods' potential to improve emissions monitoring strategies, especially as the Mediterranean region prepares for

Emission Control Area regulations in 2025.

## 1 Introduction

Maritime transport is the primary mode for European imports and exports globally, accounting for 80 % of the EU's external freight trade (Eurostat, 2023; EEA, 2018; Merico et al., 2017; EEA, 2016). Future forecasts predict a growth of the sector, with freight transport doubling in 2030 compared to 2020 (United Nations Conference on Trade and Development, 2023). While

this mode of transport is a key contributor to social and economic development worldwide (Bagoulla and Guillotreau, 2020;




Eyring et al., 2010), it negatively impacts air quality in coastal areas and global climate (Toscano, 2023; Viana et al., 2014; and European Environment Agency et al., 2013).

Ship engines are well-known sources of various pollutants as nitrogen oxides ($NO_X$), sulfur dioxide ($SO_2$), carbon monoxide (CO), volatile organic compounds (VOCs) but also greenhouse gases as carbon dioxide ($CO_2$) and methane ($CH_4$) (Aakko-Saksa et al., 2023; Celik et al., 2020; Lou et al., 2019). Globalwise, shipping accounts for approximately 15 % of $NO_X$, and 5-8 % of sulfur dioxide ($SO_2$) emissions but only to 3 % of $CO_2$ emissions (International Maritime Organization, 2020). In coastal cities shipping constitutes a major source of fine and ultrafine particle ($PM_1$ and $PM_{0.1}$)(Garcia-Marlès et al., 2024; Eger et al., 2023). These particles often contain an important fraction of toxic substances as heavy metals, polycyclic aromatic hydrocarbons (PAHs) and black carbon (BC) (Heikkilä et al., 2024; Zhao et al., 2020; Muñoz et al., 2018; Betha et al., 2016) associated to cardiovascular and respiratory diseases (Kiihamäki et al., 2024; Sofiev et al., 2018). UFPs are even more dangerous as they can deeper penetrate the pulmonary epithelium and reach various organs throughout the body (Schraufnagel, 2020). Recent research by Allouche et al. (2022) has linked pollutant exposure to weakened antiviral cellular response. Additionally, several studies suggest that long-term exposure to air pollution and living near high-traffic roadways are associated with increased risks of Alzheimer's disease, Parkinson's disease, and vascular dementia (Calderón-Garcidueñas and Ayala, 2022; Grande et al., 2020; Chen et al., 2017; Jung et al., 2015).

Approximately 70 % of ship emissions occur within 400 km of coastlines (Eyring et al., 2010) emphasizing their local impact. Moreover, the longer the average turnaround time for ships, the greater the risks for population's health and the environment (Ducruet et al., 2024). Air quality measurements in port cities reveal that ship emissions are quantitatively comparable to road transport emissions contributing substantially to air quality degradation (Tang et al., 2020; Air PACA, 2017; Viana et al., 2014; Gravgård Pedersen et al., 2013) and to severe health impacts, causing up to 0.5 % of global mortality (Mueller et al., 2023).

Since 1973, the International Maritime Organisation (IMO) has implemented the International Convention for the Prevention of Pollution from Ships (MARPOL Convention) to limit the maritime pollutant emissions under the MARPOL convention (International Maritime Organization, 2021). These regulations include restrictions on $NO_X$, $CO_2$ and $SO_2$ emissions as well as fine particles in designated Emission Control Areas (ECAs). On January 1, 2020, the IMO introduced new regulations reducing the maximum sulfur content in exhaust gases from 3.5 % to 0.5 % globally, with stricter limits of 0.1 % within Sulfur Emission Control Areas (SECA). Following negotiations that began in 2016, the Mediterranean states have jointly adopted a SECA. Called SECA Med, it will cover the entire Mediterranean by May 2025.

To comply with these new requirements, the use of exhaust gas cleaning systems has become increasingly widespread (Heikkilä et al., 2024). Exhaust Gas Recirculation (EGR) technologies recirculate a portion of the cooled exhaust gas back into the engine cylinder lowering combustion temperature and reducing $NO_X$ formation. The Selective Catalytic Reduction (SCR) systems inject urea into the exhaust stream that decomposes to ammonia promoting the catalytic conversion of $NO_X$ into nitrogen and water (Napolitano et al., 2022). These technologies enable also compliance of environmental regulations for ships running on Heavy Fuel Oil (HFO) with sulfur content exceeding 0.5 % (Laasma et al., 2022). Alternatively, more refined diesel-types fuels such as Marine Diesel Oil (MDO) or Marine Gas Oil (MGO) can be used without requiring scrubbers. The





adoption of low-sulfur fuels in maritime transport significantly reduces exposure to fine and ultrafine particle emissions (Mwase et al., 2020), contributing to an estimated global reduction of approximately 2.6 % in deaths from cardiovascular diseases and lung cancer related to $PM_{2.5}$. However, despite these advancements,, the use of these new marine fuels is projected to account for approximately 250,000 deaths annually (Sofiev et al., 2018).

In 2021, the shipping activity in Toulon began to gradually recover, particularly for ferries between France, Corsica, and Sardinia but the overall traffic levels lagged behind pre-pandemic period. Local inventories from CIGALE database (AtmoSud, 2024) estimated local $NO_X$ emissions from shipping accounting for 40% of the total emitted 1,200 tons. Primary $PM_{2.5}$ and $PM_{10}$ emissions are evaluated at 140 and 175 tons, respectively, with a contribution of shipping between 5-6 %. While $SO_2$ and $CO_2$ emissions account for 34.8 and 650,000 tons, respectively, with a maritime transport contribution of 35.2 %and 4.2

% respectively. These data underscore the significant impact of maritime activities on local air quality, necessitating a more detailed analysis of their contributions relative to other pollution sources.

    Positive Matrix Factorization (PMF) analysis of Aerosol Mass Spectrometry (AMS) measurements has been intensively used to apportion PM sources in diverse environments (Bozzetti et al., 2017; Dall'Osto et al., 2015; Zhang et al., 2011; Ng et al., 2010), but distinguishing primary sources within organic aerosol (OA) remains a key challenge in PMF. Yuan et al. (2012)

suggested that the PMF factors might reflect different stages of photochemical processing rather than entirely independent sources. In heavily polluted areas, Aiken et al. (2009) noted that AMS PMF often merges multiple sources into a single factor due to overlapping emission patterns caused by hard ionization. This observation was corroborated by Brinkman et al. (2006), who found that highly correlated sources, such as diesel and gasoline exhaust, were frequently grouped into a single PMF factor. Although PMF techniques have significantly improved source identification, accurately differentiating Hydrocarbon-

like OA (HOA) sources remains a persistent challenge using High Resolution Time-of-Flight AMS (HR-ToF-AMS) data, due to the extensive molecular fragmentation induced by the electron impact ionization, which results in overlapping mass spectral patterns and limits the resolution of HOA signatures.

    The present study offers a comprehensive insight into shipping emissions by quantitatively assessing emission factors (EFs) of $SO_2$, $NO_X$, CO, NO, $CH_4$, and particle chemical families like BC, organics (Org), sulfate ($SO_4^{2-}$), nitrate ($NO_3^-$), ammonium

($NH_4^+$), and PAHs. Additionally, particle number concentration and size distribution, in combination with PMF analysis of the non-refractory $PM_1$ (NR-$PM_1$) OA fraction are used to provide a better understanding of the impact of shipping activities on air quality in the port area. This work was able to distinguish maritime emissions from road transport emissions, providing a robust framework for investigating targeted emission sources, and may support a better air quality management and regulatory policies.

**2   Material and Methods**

**2.1   Site Presentation and Instruments**

From August 24 to September 21, 2021, a measurement campaign was conducted in the port of Toulon, a Mediterranean port city in southeastern France, as part of the AER-NOSTRUM project. The reference stations of AtmoSud and the Massalya





mobile air analysis laboratory, operated by the Laboratory of Environmental Chemistry of Marseille (LCE) and equipped with
state-of-the-art instruments (listed in Table 1), were deployed at the Toulon TCA terminal ($43°7'1''$ N, $5°56'5.2''$ E).

This location is representative of pollution directly impacting local population living in cities with important maritime activities, as the city center is located near ferry terminals and important roads (Figures S1 and S2 in the Supplement). Figure S3 indicates the arrivals and departures schedule of ferry reported by the Toulon port authority. Most departures occurred around 7 a.m. and arrivals around 10 p.m. (local time). Increased road transport was observed before arrivals and after departures of
ferries.

**Table 1.** Measurement and instrumentation deployed during the campaign.

| Measurement | Instrument, Manufacturer | Size Range | Temporal resolution |
|---|---|---|---|
| PN concentration | Envi-CPC 100, Palas | 4 nm – 5 $\mu$m | 1 s |
| $PM_1$ Particle mobility size distribution | SMPS 3936[1], TSI Inc | 15 nm – 660 nm | 2 min |
| PN and p.m. concentrations | OPC model 1.109, Grimm Aerosol Technik | 0.25 $\mu$m – 32 $\mu$m | 1 min |
| Aerosol BC concentration | AE33, Aerosol Magee Scientific | 880 nm wavelength[2] | 1 min |
| Aerosol BC concentration | MAAP 5012, Thermo Fisher Scientific Inc[3] | 637 nm wavelength | 1 min |
| Non-refractory $PM_1$ chemical composition | HR-ToF-AMS, Aerodyne Research Inc | 70 nm – 700 nm[4] | 30 s |
| $SO_2$ concentration | API100E, Teledyne Technology Inc | - | 10 s |
| $NO_X$, NO, $NO_2$ concentrations | API200E, Teledyne Technology Inc | - | 10 s |
| $CO_2$, CO, $CH_4$ concentrations | G2401, Picarro Inc | - | 5 s |
| Wind speed, wind direction | Tridi USA-1, METEK | - | 10 s |

[1] Coupled with CPC model 3775, Classifier model 3080 and DMA model 3081 from TSI Inc.

[2] AE33 measured wavelengths are 370, 470, 520, 590, 660, 880 and 950 nm, but only 880 nm wavelength data were used in this paper.

[3] Used only for CE calculation.

[4] Size range of transmission efficiency of aerodynamic lens system of Aerodyne HR-ToF-AMS (Liu et al., 2007).

The distribution of submicron particles between 15 and 661.2 nanometers, across 106 size channels, was measured using a Scanning Mobility Particle Sizer (SMPS). The measurements were conducted with a 2-minute time step using a TSI model 3936 SMPS, which combines a 3080 Electrostatic Classifiers with a Classifiers with a Differential Mobility Analyzer (DMA 3081), a Condensation Particle Counter (CPC 3775), and a $^{85}$Kr neutralizer. $CO_2$, CO and $CH_4$ was measured by Picarro
analyzer, calibrated according to the procedure described in Xueref-Remy et al. (2023). Black carbon was sampled using a Thermo Fischer multi-angle absorption photometer (MAAP) and a Aerosol Magee Scientific aethalometer AE33 with a 1-minute time step for both instruments. The non-refractory submicron fraction of aerosol was continuously measured using a HR-ToF-AMS. A detailed description of the HR-ToF-AMS is available in Canagaratna et al. (2007); DeCarlo et al. (2006). Data analysis was performed using the HR-ToF-AMS analysis software Squirrel version 1.65B and Pika version 1.25B, based on
high-resolution fitting procedures outlines in DeCarlo et al. (2006). Calibration of the instrument in brute-force single-particle mode was carried out before the campaign using ammonium nitrate and ammonium sulfate, yielding a nitrate Ionization Efficiency (IE) of $5.07 \times 10^8$ and Relative Ionization Efficiencies (RIE) of 3.91 for ammonium and 1.7 for sulfate. Default RIE



values of 1.1, 1.3, and 1 were applied for nitrate, chloride, and organic fractions, respectively (Xu et al., 2018; Canagaratna et al., 2007). Instrument resolution was set to 30 seconds in V-mode, with high-resolution analysis conducted for *m/z* ratios ranging from 12 to 256. The SMPS, MAAP, and HR-ToF-AMS sampled air from the same line, and an average Collection Efficiency (CE) of 0.63 was calculated to correct HR-ToF-AMS concentrations. During intense ship plume events, a unit CE value was determined based on comparisons between SMPS, HR-ToF-AMS and MAAP measurements, a behavior also reported in the literature (Voliotis et al., 2021; Quinn et al., 2006). The standard deviation of the CE was estimated at 14 %, consistent with the 20 % uncertainty reported in previous studies (Bahreini et al., 2009; Brendan M. Matthew and Onasch, 2008).

## 2.2 Emission Factors and Fuel Sulfur Content Calculations

To determine the EFs of pollutants from shipping, a carbon mass balance approach is used. This method involves measuring pollutant concentrations, particularly $CO_2$, at a receptor site, i.e., a location where ship plumes intersect. The measured concentrations include both atmospheric background and pollution introduced by the plume. We use linear fit-based EFs that linearly interpolate background concentrations between the levels before and after the plume (Volent et al., 2025; Diesch et al., 2013). This technique for assessing background concentration specifically addresses the Toulon area, where other sources affect the accurate evaluation of background concentration across an extended period. The background has been estimated from measurements taken before and after the plume event. The concentrations of various pollutants within the plume are then correlated to fuel consumption which is quantified based on the plume's $CO_2$ concentration (Celik et al., 2020; Ausmeel et al., 2019; Ježek et al., 2015; Lack et al., 2009).

$$EF_x = \frac{\int_E^G ([x]_G(t) - [x]_E(t))dt}{\int_E^G ([CO_2]_G(t) - [CO_2]_E(t))dt} \times \frac{M_{CO_2}}{M_C} \times \omega_c \tag{1}$$

Where $EF_x$ represents the emission factor of substance $X$ expressed in grams of pollutant emitted per kilogram of fuel consumed (g/kg$_{fuel}$), the constant term $M_{CO_2}/M_C$ corresponds to the inverse of the mass fraction of carbon in $CO_2$, $\omega_c$ denotes the mass fraction of carbon in ship fuel, $[x]$ is the excess concentration of the substance $x$ after subtracting the background level, expressed in particle per cubic meter or $\mu g$ per cubic meter, and $[CO_2]$ is the excess concentration of $CO_2$ after background subtraction, expressed in $mg/m^3$. $E$ and $G$ mark the start and the end of the plume, respectively. Since most ferries are powered by diesel engines, the $\omega_c$ value has been set to 0.865 kg of carbon per kg of fuel, corresponding to the mass fraction of carbon in marine diesel fuel (Diesch et al., 2013).

The fuel sulfur content (FSC) is derived from the ratio of excess $SO_2$ to $CO_2$ concentrations in the plume, assuming complete conversion of fuel sulfur to $SO_2$ (Volent et al., 2025; Van Roy et al., 2022; Pirjola et al., 2014). This yields the following expression:

$$FSC(\%) \approx EF_{SO_2} \times \frac{M_C}{M_{CO_2} \times \omega_c} \times 0.232 \tag{2}$$



Here, $FSC$ is the fuel sulfur content in %, $EF_{SO_2}$ is the sulfur emission factor in (g/kg$_{\text{fuel}}$). This method provides a lower-limit estimation of FSC, as a small fraction of sulfur may be emitted as $SO_3$ or converted into $H_2SO_4$ in the atmosphere (Pirjola et al., 2014; Alföldy et al., 2013; Moldanová et al., 2013).

### 2.3 Theta angle

The theta angle, or cosine similarity, is a method increasingly used to calculate correlation when comparing mass spectra (Bougiatioti et al., 2014; Kostenidou et al., 2009). A mass spectrum with a dimension $n$ (representing the number of *m/z* fragments that compose the mass spectrum) and $\alpha_i$ the intensity of the *m/z$_i$* fragment, is treated as an n-dimensional vector. Thus, a mass spectrum $\mathbf{A}$ can be expressed as:

$$\mathbf{A} = \alpha_1 m/z_{\mathbf{1}} + \alpha_2 m/z_{\mathbf{2}} + ... + \alpha_n m/z_{\mathbf{n}} \tag{3}$$

Then, the cosine between two mass spectra $\mathbf{A}$ and $\mathbf{B}$ can be calculated:

$$cos(\theta) = \frac{\mathbf{A} \times \mathbf{B}}{\|\mathbf{A}\| \times \|\mathbf{B}\|} \tag{4}$$

Bougiatioti et al. (2014) define the similarity of two mass spectra as follow: a theta angle between 0 and 15° indicates that the two mass spectra are similar, an angle between 15 and 30° suggests a weak correlation, and an angle greater than 30° indicates that the two mass spectra are different. However, it is important to note that even for theta angles exceeding 30°, valuable insights into potential associations between mass spectra can still be obtained. That's why Kostenidou et al. (2009) assumes $cos(\theta)$ is analogous to Pearson's coefficient correlation (R) when comparing mass spectra.

### 2.4 Spectral Relative Predominance

A Spectral Relative Predominance (SRP) is a metric developed to evaluate the relative differences in ion intensities between two mass spectra, highlighting which mass spectrum predominantly produces specific ions. For two mass spectra $\mathbf{A}$ and $\mathbf{B}$, that can be expressed as a $n$-dimensional vector as in equation (2), with intensities $\alpha_i$ and $\beta_i$ for ion $i$, the SRP is defined as:

$$\text{SRP}_i = \begin{cases} \frac{\alpha_i - \beta_i}{\beta_i} & \text{if} \alpha_i > \beta_i, \\ -\frac{\beta_i - \alpha_i}{\alpha_i} & \text{otherwise.} \end{cases} \tag{5}$$

Positive SRP values indicate that ion $i$ is predominantly produced in mass spectrum $\mathbf{A}$, with the magnitude reflecting the proportional increase relative to mass spectrum $\mathbf{B}$, whereas negative values signify predominance in mass spectrum $\mathbf{B}$, scaled by the proportional increase over $\mathbf{A}$. This signed, asymmetric measure offers a quantitative tool to assess ion-specific differences, providing insights into the comparative contributions of ions across mass spectra, contrasting with the theta angle that considers the whole spectra.

### 2.5 Positive Matrix Factorization

The PMF model developed by Paatero and Tapper in 1994 is an analytical tool based on decomposing a positive matrix $\mathbf{X}$, into two non-negative matrices, $\mathbf{G}$ and $\mathbf{F}$, such that their product best approximates the original matrix while minimizing the



residual matrix $\mathbf{E}$:

$$\mathbf{X} = \mathbf{G} \times \mathbf{F} + \mathbf{E} \tag{6}$$

where $\mathbf{X}$ is a $n \times m$ matrix representing chemical concentration measurements at different time points $m$ and for different chemical species $n$, $\mathbf{G}$ is a $n \times p$ matrix, where $p$ is the number of potential profiles. Each column of $\mathbf{G}$ represents the temporal concentration series o f a factor. $\mathbf{F}$ is a $p \times m$ matrix describing the chemical profiles of the factors. $\mathbf{E}$ is a $n \times m$ matrix representing the difference between $\mathbf{X}$ and the product $\mathbf{G} \times \mathbf{F}$. Minimizing the residual matrix $\mathbf{E}$ constitutes a fundamental aspect in solving equation (4), where the model endeavors to optimize the function $\mathbf{Q}$:

$$\mathbf{Q} = \sum_i \sum_j \left( \frac{e_{i,j}}{\sigma_{i,j}} \right)^2 \tag{7}$$

where $e_{i,j}$ represents the model residuals for species $j$ at time $i$ and $\sigma_{i,j}$ represents the uncertainty for species $j$ at time $i$. It is noteworthy that there are no unique solutions for a given value of $\mathbf{Q}$, and a lower value of $\mathbf{Q}$ does not necessarily lead to a better deconvolution. This drawback may be caused by rotational ambiguity. To mitigate this ambiguity, it is possible to constrain the matrices $\mathbf{F}$ and/or $\mathbf{G}$ with external constraints. To reduce rotational ambiguity, the ME-2 solver has been developed (Paatero, 1999), allowing the constraint of chemical profiles or temporal evolution of factors, notably with an "a-value" approach, i.e. a degree of freedom (defined by the scalar a), corresponding to the extent to which the factor profile can deviate from the provided constraint. This approach also helps to avoid unrealistic solutions and enables the separation of sources with similar chemical signatures (Canonaco et al., 2013; Lanz et al., 2008).

## 3 Results and Discussions

### 3.1 Campaign Overview

An overview of the measurements campaign is provided in Figure 1, including meteorological data (wind speed and direction) HR-ToF-AMS species and black carbon, as well as PN, PM and gases ($NO_X$ and $SO_2$). The pie chart representing the median $PM_1$ chemical composition shows the following proportions: 53 % organics, 16 % sulfate, 7 % ammonium, 2 % nitrate and 21 % BC. The most intense $PM_1$ peaks are associated with ship arrivals and departures when the wind originates from the sea, with directions ranging from 130 to 290° (northwest to east). Shipping plumes are associated with high concentrations of PN, PM (Org, BC) and $NO_X$. These intense plumes peaks are not accompanied by a significant increase in sulfate, which appears to be primarily driven by regional background levels.

The diurnal profiles of the measured pollutants are shown in Figure S4 in the Supplement, where two distinct patterns can be observed. The first is linked to the change in breeze direction, where the onshore breeze leads to high concentrations of gas tracers such as CO, $CO_2$ and $CH_4$. In contrast, although to a lesser extent, $SO_2$, particle matter (PN and PM), Org and BC, show higher levels during ship arrivals and departures, particularly between 7-9 a.m. and 6-9 p.m. (local time), emphasizing the port's significant contribution to local pollution. Table S1 in the Supplement provides a breakdown of vessel types and their





respective proportions, with ferries constituting the majority of the fleet (78.5 %), followed by smaller shares of cruise ships, tankers, yachts, and other vessel categories, which together account for 17.1 %. 4.4 % of the vessels could not be identified.

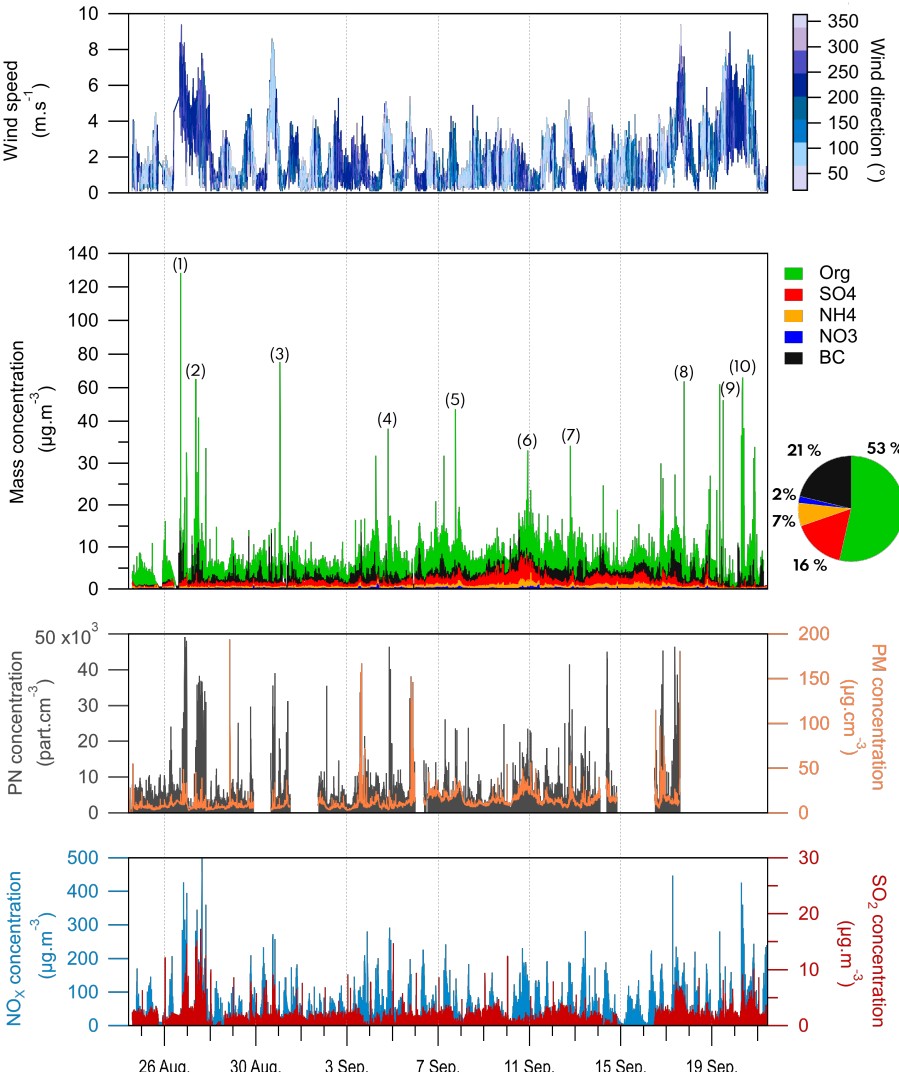

**Figure 1.** Concentrations of organics, nitrate, sulfate, ammonium, chloride, BC, PN, PM, $NO_X$, $SO_2$ and wind speed colored with wind direction, unit CE applied to HR-ToF-AMS data. Pie-chart represents mass contribution of $PM_1$. The numbers above the organic peaks indicates the peaks used in section 3.2.3.





## 3.2 Emission factors

EFs are commonly used to evaluate the contribution of different sectors to local air quality. A real challenge is distinguishing emissions from different urban sources, particularly those from road transport and shipping. An effective approach to achieve this differentiation is by examining the wind direction to trace the origin of pollution plumes; if plumes come from the sea, they are most likely attributed to maritime sources, specifically ships. When particle number concentration peak exceeds twice the average background level and originates from the sea is attributed to a ship plume. To distinguish offshore events from those originating on land, a selection criterion is applied: only events with an average wind direction between 130° and 290° (southeast to northwest) and a standard deviation of less than 50° are considered. The beginning and end of each plume are determined based on PN and $CO_2$ concentration data, adjusted according to the instrumental time steps.

Out of 69 detected plumes, ferries contributed to the largest fraction with 63 % (45 plumes), followed by cruise ships and yachts, each accounting for 6 % (4 plumes), while a significant 25 % (18 plumes) were classified as "Unknown", indicating ships that could not be accurately identified. Some plumes were attributed to specific ferries, and ferries identified multiple times during arrivals and departures were labeled A, B, C, D, and E. Their EFs and characteristics (tier, number of engines, fuel types, tonnage) are listed in Tables S2 and S3. Figures 2, 3 and 4 present the EFs for major gaseous pollutants ($SO_2$, $NO_X$, CO and $CH_4$), PN, and particulate pollutants (BC, Org, PAHs and $SO_4^{2-}$), respectively. The figure also compares the different EFs across different vessel types, and operation mode. Table 2 provides EFs for all identified ship plumes of gaseous ($SO_2$, $NO_X$, CO, NO, $CH_4$), particulate (Org, $NO_3^-$, $NH_4^+$, $SO_4^{2-}$ BC and PAHs) and PN concentrations.

**Table 2.** Emission factors of all identified ship plumes. $SO_2$, $NO_X$, CO, NO, $CH_4$, Org, $SO_4^{2-}$ and BC in g/kg$_{fuel}$. $NO_3^-$, $NH_4^+$ and PAHs in mg/kg$_{fuel}$) and PN in particle/kg$_{fuel}$.

| All EFs | $SO_2$ | $NO_X$ | CO | NO | $CH_4$ | Org | $SO_4^{2-}$ | $NO_3^-$ | $NH_4^+$ | BC | PAHs | PN |
|---|---|---|---|---|---|---|---|---|---|---|---|---|
| Median | 0.27 | 14.26 | 20.58 | 5.92 | 0.99 | 1.04 | 0.08 | 14.9 | 29.9 | 0.28 | 6.0 | 3.44E+15 |
| Mean | 0.45 | 20.74 | 23.03 | 8.10 | 1.31 | 1.73 | 0.13 | 21.6 | 67.6 | 0.38 | 10.3 | 4.81E+15 |
| 1st Quartile | 0.09 | 9.26 | 13.91 | 2.66 | 0.34 | 0.41 | 0.03 | 5.1 | 17.3 | 0.17 | 2.9 | 1.45E+15 |
| 3rd Quartile | 0.61 | 26.37 | 27.90 | 10.88 | 1.63 | 1.92 | 0.17 | 25.6 | 73.3 | 0.45 | 11.4 | 7.45E+15 |
| Number | 62 | 55 | 57 | 66 | 63 | 61 | 50 | 52 | 62 | 66 | 60 | 55 |

### 3.2.1 Sulfur Dioxide

The mean FSC of ship fuels is 0.03 %, below the IMO2020 requirements of 0.5 %, the mean $SO_2$ EF is 0.45 g/kg$_{fuel}$, significantly lower than most of the reported values before the implementation of the IMO2020 regulation. This is nearly 50 times lower than reports from Celik et al. (2020) in the Mediterranean Sea and around the Arabian Peninsula (26±6 g/kg$_{fuel}$), 17 times lower than those of Diesch et al. (2013) in the Elbe River in Northern Germany (7.7±6.7 g/kg$_{fuel}$) and 6 times lower than results from MDO fueled tanker with 0.1 % of mass sulfur content (2.9±0.2 g/kg$_{fuel}$) Sinha et al. (2003). Nevertheless, the $SO_2$





emission factor observed in Toulon is comparable to recent values, such as 0.4 g/kg$_{fuel}$ in Marseille (Le Berre et al., 2024), and 1.5 g/kg$_{fuel}$ on a MGO-powered ferry (Timonen et al., 2022).

Remarkably, the mean SO$_2$ EF is lower for cruise ships, 0.13 g/kg$_{fuel}$, likely due to the use of exhaust gas cleaning systems, as indicated on cruise line websites. The yachts identified are equipped with engines designed to run on ultra-low sulfur diesel (ULSD), in agreement with a mean SO$_2$ EF of 0.28 g/kg$_{fuel}$. Examining the arrivals and departures times of ferries reveals a high variability in EF across the different vessels. Notably, Ferry C exhibits significantly lower emissions compared to the other ferries, possibly due to the less powerful auxiliary engines (3 × 1,680 kW, see Table S3), while ferries A, B and D have more powerful auxiliary engines (more than 6,000 kW) and exhibit the highest EF. The recent calculated SO$_2$ EF in Toulon and Marseille are considerably lower than the regional inventory with 2 g/kg$_{fuel}$, highlighting the need for continuous measurements to adjust regional inventories in response to regulatory changes and fleet evolution.

### 3.2.2 Nitrogen oxides

The mean NO$_X$ EF is of 20.7 g/kg$_{fuel}$, representing a reduction of approximately 44–68 % compared to the values documented by Le Berre et al. (2024) (median 37 g/kg$_{fuel}$), Celik et al. (2020) (mean 51 g/kg$_{fuel}$), Betha et al. (2016) (51-64 g/kg$_{fuel}$ for ULSD), and Winnes et al. (2016) (64.5 g/kg$_{fuel}$ for MGO). This decrease may reflect a combination of factors, including engine optimization, engine loads and operational practices (Sugrue et al., 2022; Grigoriadis et al., 2021; Peng et al., 2020). The NO$_X$ EF from ferries show significant variability, ranging from 5.2 g/kg$_{fuel}$ to 72.5 g/kg$_{fuel}$, with a mean of 25 g/kg$_{fuel}$, and aligning with Zhang et al. (2024) findings with median values of 22.3 g/kg$_{fuel}$ for auxiliary engines at 50 % load in ferry operations. During ferries maneuvers (arrivals and departures), auxiliary engines are more frequently used, contributing to higher NO$_X$ emissions, especially during departures, as cold engines lead to a less efficient combustion. Only ferry C presented an outlier of 60 g/kg$_{fuel}$ during one arrival, possibly linked to its lower auxiliary power (3 × 1,680 kW) and specific conditions as higher engine load. Cruise ships exhibit the lowest EF, mean 15.3 g/kg$_{fuel}$, likely due to the use of catalytic converters. The actual regional inventory of NO$_X$ from shipping are of 80.0 g/kg$_{fuel,}$ highlighting again, as for SO$_2$, the need of updated measurements.

### 3.2.3 Carbon oxide

The mean CO EF is 23 g/kg$_{fuel}$, consistent with Celik et al. (2020) (20±3 g/kg$_{fuel}$), reflecting typical emissions from ship engines using fuels like low-sulfur HFO (LSHFO), very low-sulfur fuel oil (VLSFO), MGO, and MDO. However, this mean is approximately four times higher than the median EF reported in Marseille by (Le Berre et al., 2024) (5.4 g/kg$_{fuel}$) and three times higher than regional inventories (7.5 g/kg$_{fuel}$). A difference likely stemming from operational modes, including low-speed maneuvers required for docking, as low engine-loads can increase CO emissions as reported by (Bai et al., 2020). The EF of CO vary from 2.43 g/kg$_{fuel}$ to 57.87 g/kg$_{fuel}$ (median of 20.6 g/kg$_{fuel}$) underscoring how factors like engine start-up/shutdown, incomplete combustion and auxiliary engines use during port operations highly influence the CO emissions. Aside from outliers — lowest value of 2.43 g/kg$_{fuel}$ for ferry A and highest value of 57.87 g/kg$_{fuel}$ for ferry C— the EF of CO show minimal differences between arrivals and departures, suggesting stable combustion under typical operating conditions, though engine design (Tier I for D, Tier II for A, B, C, E) and fuel type (LSHFO, VLSFO, MGO, MDO).



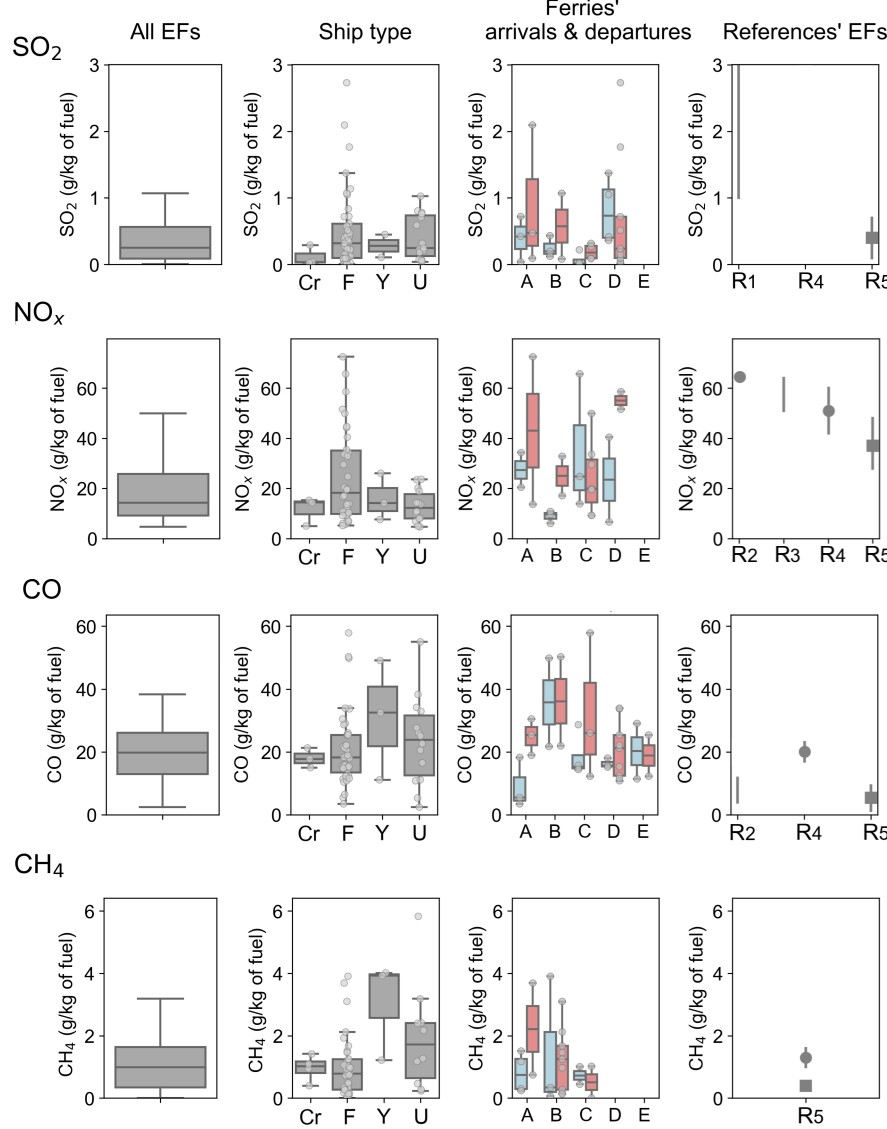

**Figure 2.** Emission factors of major gaseous pollutants (gray dots) for all vessels and for specific categories (Cr, F, Y and U refer to cruiseships, ferries, yachts and unknown, respectively). A, B, C, D, and E refer to specific ferries which arrivals (light-blue colored) and departures (light-coral colored) are depicted. In references EFs, square represents median while dots refer to mean EFs with vertical line representing standard deviation or first-to-third quartiles. R1, R2, R3, R4 and R5 stands for Diesch et al. (2013), Betha et al. (2016), Winnes et al. (2016), Celik et al. (2020) and Le Berre et al. (2024), respectively.





### 3.2.4 Methane


The median $CH_4$ EF of 1.0 g/kg$_{fuel}$ is slightly higher than values reported from Marseille in 2021 of median 0.4 g/kg$_{fuel}$ (Le Berre et al., 2024) and similar to reports from Volent et al. (2025) (median of 0.99 g/kg$_{fuel}$). This EF is considerably higher than other studies with EFs of 0.02 g/kg$_{fuel}$ and 0.05 g/kg$_{fuel}$ from (Cooper, 2003; Timonen et al., 2022),respectively. Among the ships identified none was LNG-fueled, but the yachts exhibited highest $CH_4$ emissions with values up to 3.9 g/kg$_{fuel}$. Methane

emissions arise from the incomplete combustion of hydrocarbons in fuels and depend on fuel composition, engine design and possible slip of unburned fuel occurring at low speed or idle in untuned engines (Penman et al., 2001). Inadequately tuned engines, such as those on yachts tend to emit much more methane, in line with the emissions for small vessels as previously reported by Wang et al. (2022) (5.2 g/kg$_{fuel}$).

### 3.2.5 Particle number

The mean EF of PN (Figure 3), measured with a CPC, is approximately $4.8 \times 10^{15}$ part/kg$_{fuel}$ a value slightly lower than those found in the literature. This EF is consistent with previous reports for vessels using low-sulfur fuels by Betha et al. (2016) (8.0 $\times 10^{15}$ to $1.1 \times 10^{16}$ part/kg$_{fuel}$ for ultra-low sulfur diesel) and Le Berre et al. (2024) (median of $6.7 \times 10^{15}$ part/kg$_{fuel}$), possibly reflecting a decrease in particle formation due to reduced sulfur fuel content. The highest PN EF of $1.63 \times 10^{16}$ part/kg$_{fuel}$ is observed during departures of ferries. However, no significant difference is observed between arrivals and departures for the

same ferry, suggesting stable particle emissions under different operational conditions, though engine design and fuel quality play a critical role.

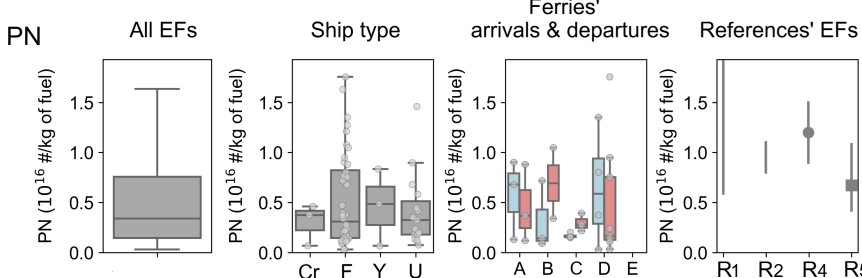

**Figure 3.** Emission factors of particle number (gray dots) for all vessels and for specific categories (Cr, F, Y and U refer to cruiseships, ferries, yachts and unknown, respectively). A, B, C, D, and E refer to specific ferries which arrivals (light-blue colored) and departures (light-coral colored) are depicted. In references EFs, square represents median while dots refers to mean EFs with vertical line representing standard deviation or first-to-third quartiles. R1, R2, R3, R4 and R5 stand for Diesch et al. (2013), Betha et al. (2016), Winnes et al. (2016), Celik et al. (2020) and Le Berre et al. (2024), respectively.





### 3.2.6 Black carbon

The mean CO EF of 0.38 g/kg$_{fuel}$ is consistent with literature reported in Marseille in 2021(Le Berre et al., 2024) (0.48 g/kg$_{fuel}$ for maneuvering ships) and with a cargo vessel (0.48 g/kg$_{fuel}$) (Huang et al., 2018) and reflects typical emissions from diesel
marine engines using low-sulfur fuels. This CO EF, ranging from 0.03 g/kg$_{fuel}$ to 1.90 g/kg$_{fuel}$ (median of 0.28 g/kg$_{fuel}$), show small variation across ship types or vessel size. Stable BC EF post-IMO 2020 suggest that BC is primarily driven by engine load and the type of combustion as lean/rich regimes (Le Berre et al., 2024; Mueller et al., 2023). Consequently, BC remains of major interest due to its significant contribution to air pollution, climate and possible adverse health effects.

### 3.2.7 Organic and polycyclic aromatic hydrocarbons

The mean and median Org EF are 1.73 g/kg$_{fuel}$ and 1.04 g/kg$_{fuel}$, respectively, in quite good agreement with literature values reported by Le Berre et al. (2024) (median 0.86 g/kg$_{fuel}$),Celik et al. (2020) (mean 3.0 g/kg$_{fuel}$), and Diesch et al. (2013) (mean 1.8 g/kg$_{fuel}$). The EF range from 0.13 g/kg$_{fuel}$ to 12.1 g/kg$_{fuel}$, this variability can tentatively be explained by engine type, fuel quality, and operational conditions, particularly for ferries. The Org EFs are generally higher during departures than arrivals for all identified ferries. The highest values are observed for ferries A, B and D characterized by powerful auxiliary engines
(more than 6,000 kW cumulated for each ferry).

The EF of PAHs, corresponding to the sum of PAHs families defined by Herring et al. (2015), exhibits a mean value of 10.3 mg/kg$_{fuel}$, aligning closely with values from Celik et al. (2020) (mean 11 mg/kg$_{fuel}$) and Diesch et al. (2013) (mean 5.3 mg/kg$_{fuel}$). As for the EF of Org, the EF of PAHs is generally higher during departures than arrivals with peaks reaching 60 mg/kg$_{fuel}$ (e.g., for ferries with higher engine loads). This pattern underscores the impact of operational practices, such as cold
engine, low-speed operations and fuel switching.

### 3.2.8 Sulfate

The mean EF for SO$_4^{2-}$ is 0.13 g/kg$_{fuel}$, reflecting a significant decreased compared to previous reports, being approximately 30 times lower than the 4 g/kg$_{fuel}$ value reported by (Celik et al., 2020) and 4 times lower than that of Diesch et al. (2013) (0.54 g/kg$_{fuel}$). This reduction is consistent with more recent findings of Le Berre et al. (2024) (median of 0.05 g/kg$_{fuel}$). This
decrease reflects the impact of reduced sulfur content in marine fuels. Similar to the Org and SO$_2$ EF, SO$_4^{2-}$ EF is generally higher during departures than arrivals for all identified ferries, with values ranging from 0.003 g/kg$_{fuel}$ to 0.65 g/kg$_{fuel}$, likely due to increased engine loads, higher sulfur residuals in fuel during high-power operations, and use of auxiliary engines during maneuvers. This variability underscores the influence of operational practices on sulfate emissions, highlighting the need for further analysis of engine design and fuel management strategies to maintain low SO$_4^{2-}$ levels.




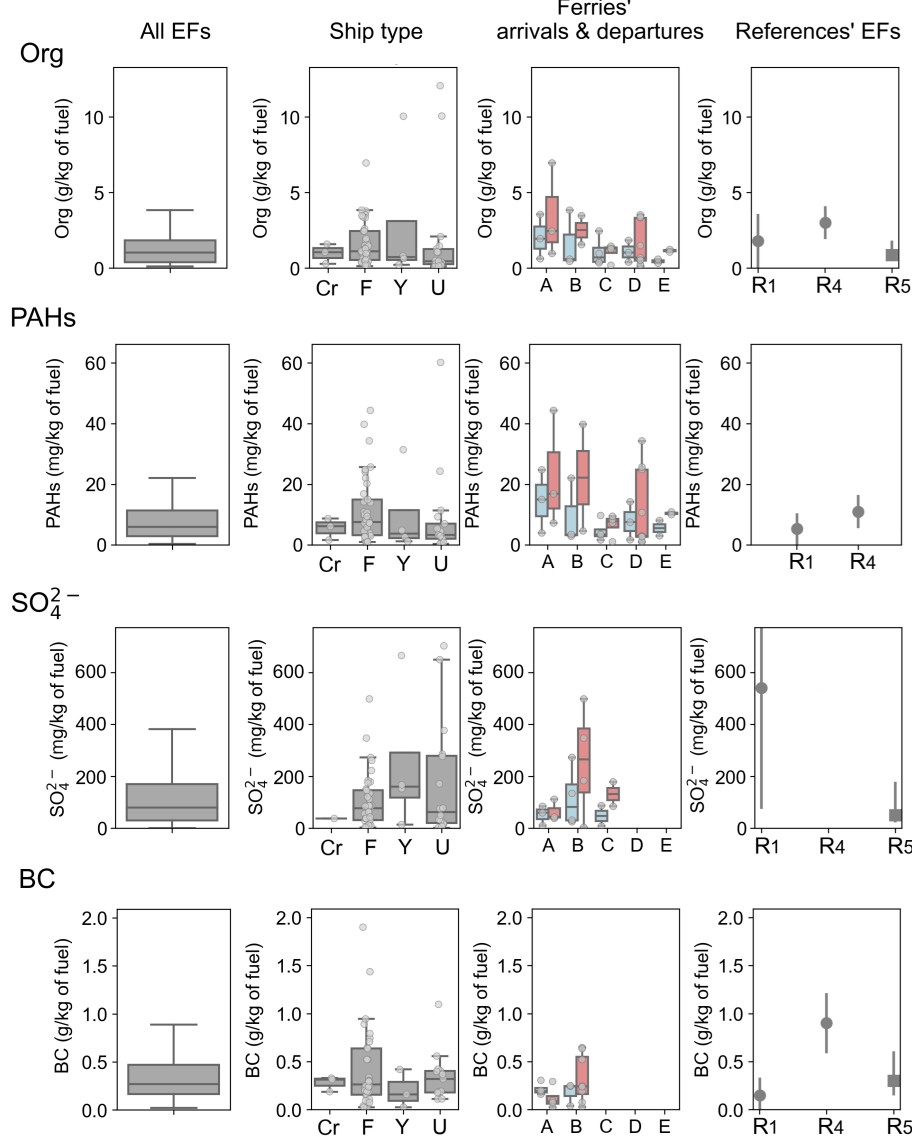

**Figure 4.** Emission factors of major particulate pollutants (gray dots) for all vessels and for specific categories (Cr, F, Y and U refer to cruiseships, ferries, yachts and unknown, respectively). A, B, C, D, and E refer to specific ferries which arrivals (light-blue colored) and departures (light-coral colored) are depicted. In references EFs, square represents median while dots refers to mean EFs with vertical line representing standard deviation or first-to-third quartiles. R1, R2, R3, R4 and R5 stand for Diesch et al. (2013), Betha et al. (2016), Winnes et al. (2016), Celik et al. (2020) and Le Berre et al. (2024), respectively.





### 3.2.9 Key insights and implications

The analysis of emissions as a function of the operational phase shows that the levels of some pollutants as sulfates, organics and PAHs and $NO_X$ are higher during departures, reflecting the impact of cold engine impact, incomplete combustion and low engine loads. The majority of identified ferries operate on marine diesel oil (MDO) or marine gas oil (MGO) and low-sulfur fuels (LSHFO or VLSFO), complying with sulfur limits (below 0.5 % for LSHFO/VLSFO and <0.01 % for MDO/MGO). Fuel changes after 2021 and dual-fuel usage significantly influenced EFs. Only cruise ships were equipped with scrubbers and were associated to lower $SO_2$, $NO_X$, and $SO_4^{2-}$ EFs. The highest $PM_1$ EF is related to organics, highlighting their dominant role in shipping's particulate pollution, particularly during departures. Unlike sulfur-related emissions ($SO_2$, $SO_4^{2-}$), which have decreased post-IMO 2020 due to low-sulfur fuels, organic EF remains similar to pre-regulation levels (Celik et al., 2020; Diesch et al., 2013), underscoring the importance of positive matrix factorization (PMF) analysis on organic fractions to better understand shipping's air quality impact in coastal cities.

## 3.3 PMF optimization

### 3.3.1 Input matrix and error weighting

The PMF model was populated with organic aerosol data from the HR-ToF-AMS, covering 290 compounds with $m/z$ ratios ranging from $m/z$ 12 to 256 (including 218 compounds between 12 to 150, and 72 PAHs from alkylated PAHs (APAHs), unsubstituted (non-functionalized) PAHs (UnSubPAHs) and oxygenated PAHs (OPAHs) families). For the sake of readability, ion masses presented have been rounded to the nearest nominal mass. The list of full $m/z$ masses and ions are listed in Table S4. The selected PAHs included parent ion $[M]^+$ and the associated $[M-H]^+$ ion following the identification method described by Herring et al. (2015) and Dzepina et al. (2007).

The dataset consisted of 37,562 time points with a one-minute time step. This resolution allows for the distinction of organic peaks associated with ship emissions, typically lasting only a few minutes. Historically, HR-ToF-AMS two-dimensional matrix is exported by Pika software with unit CE and without RIE applied; therefore, the CE correction was then applied to the PMF results. Ancillary instruments, were used to assess correlations between various measured species and the PMF solutions to identify different pollution sources.

The error matrix was down-weighted using a cell-wise signal-to-noise ratio (Brown et al., 2015), calculated at each time step to account for the fugacity of ship plumes, which are typically sampled for only a few minutes. Variables derived from $m/z$ 44 ($CO_2^+$) were excluded from the PMF analysis ($m/z$ 16, 17, 18, and 28) and reintroduced during post-processing.

As shown in Figure 1, the organic peaks are rarely associated to sulfate enhancement, as this later seems to be rather influence by background levels. To test this hypothesis, a preliminary PMF analysis was conducted, incorporating eight sulfur-containing ion fragments ($SO^+$, $SO_2^+$, $HSO_2^+$, $SO_3^+$, $HSO_3^+$, $H_2SO_3^+$, $SO_4^+$, $HSO_4^+$, $H_2SO_4^+$) into the initial input matrix. Five unconstrained PMF runs were performed, testing factors solutions ranging from five to nine, to determine if any factors were associated to these sulfate fragments.





Contrary to the findings of Fossum et al. (2024) at Dublin Port, the model did not identify any sulfate-rich ship emissions factors. In fact, in the nine-factor solution, 96 % of the sulfate signal is associated to secondary factors (displayed in Figure S5). As a result, sulfate was not further considered in PMF analysis.

### 3.3.2 PMF constraints

To better identify the optimal mass spectral combinations for representing shipping emission sources, we developed a method based on a combination of mass spectra of the highest organic peaks observed from ships (numbered in Figure 1). A multi-linear regression model was employed to assess how accurately the measured mass spectra could be reconstructed through a linear combination. The model's performance was evaluated based on the Pearson $R^2$ score, which quantifies the proportion of variance explained, and the theta angle, which measures the deviation between the true and predicted mass spectra in degrees. Combinations were considered optimal if they achieved a $R^2$ score of at least 0.95 and an angular distance of 5 degrees or less. Since the shipping-related mass spectra were selected based on the most intense ship plumes, background noise influence was minimized. The results demonstrated that a combination of three specific mass spectra provided the best overall fit, with an average $R^2$ of 0.99 and an average angular distance of 1.20 degree. Consequently, these three mass spectra were selected as constraints for shipping emissions in the PMF analysis (Figure S6).

Two additional constraints were applied: a HOA mass spectrum from Hayes et al. (2013), measured using a HR-ToF-AMS in Pasadena in 2010, and a COA mass spectrum from Elser et al. (2016), measured in China using a HR-ToF-AMS. There was no significant difference between the HOA mass spectrum measured in Pasadena in 2010 and other mass spectra measured in Europe more recently (theta angle greater than 0.93 during a campaign conducted in the center of Rome in spring 2014 (Struckmeier et al., 2016). The mass spectrum from Hayes et al. (2013) was chosen because it included PAHs ions that are used in our PMF analysis. In total, five mass spectra were used to constrain the PMF analysis, corresponding to the three ships mass spectra, named Shipping 1, Shipping 2, Shipping 3, one HOA and one COA. The full list of the constrained factor and related ions used for the PMF is available in Table S4.

### 3.3.3 Number of factors

A common approach for determining the optimal number of factors $n$ in the PMF analysis is to examine the variation in the $\mathbf{Q/Q_{exp}}$ ratio and select the solution that exhibits the most pronounced change compared to the run with $n-1$ factors. A significant change in the quantity $\mathbf{Q/Q_{exp}}$ indicates a substantial decrease in residuals and enhances the explained variability of the model. To determine the appropriate number of factors, five runs were conducted with factors numbers ranging from 6 to 10, with the aforementioned constraints. The resulting $\mathbf{Q/Q_{exp}}$ values for solutions with six, seven, eight, nine, and ten factors were 1.92, 1.33, 1.20, 1.14 and 1.09 respectively. Thus, a minimum of seven factor is suitable for PMF analysis. However, upon closer examination of the eight-factor solution, an additional factor was identified. This factor, named OxHOA (Oxidized HOA), will be discussed in section 3.4 after the interpretation of other factors, to facilitate comparative analysis.



### 3.3.4 Sensitivity analysis of the a-values

A sensitivity analysis was conducted by scanning possible a-values for all the constraints factors. For the three shipping con-
straints, a-values ranging from 0 to 0.3 with a step of 0.1 were examined (Drosatou et al., 2019; Chen et al., 2022), finally the
selected a-values were 0.2, which allowed recovery of the most intense and the highest number of plumes observed.

Since HOA and COA sources tended to mix, a sensitivity analysis of the a-value was also carried out with a-values from 0
and 1 with a step of 0.1 (Wang et al., 2024; Chazeau et al., 2022; Bozzetti et al., 2017; Elser et al., 2016), corresponding to
121 different a-value combinations. These combinations were categorized based on the identification of road transport plumes,
specifically between 8 a.m. and 9 a.m. and between 12 p.m. and 1 p.m. using correlations with external tracers as BC, $NO_X$,
and CO. Finally, the solution with a-values of 0.3 for both mass spectra has been retained.

### 3.3.5 Residuals and factor uncertainties

Rotational ambiguity was explored using a bootstrap approach (Efron, 1979). Bootstraps randomly resample time points from
the input matrix to generate a new matrix, which is then subjected to PMF analysis with the selected constraints. A number of
100 bootstrap runs were performed to estimate the uncertainty associated with the PMF solutions.

The residuals of the bootstrap average solution were examined to identify any significant deviations from the mean, which
could indicate systematic model overestimation or underestimation. Scaled residual values typically ranging between -3 and
3 confirm the validity of the PMF (Canonaco et al., 2021; Paatero and Hopke, 2003). In our case, only 1.7 % of residuals
fall outside this interval (Figure S7). Typically, the uncertainties for each factor are defined as the center of the lognormal
distribution of variability across time points (Canonaco et al., 2021; Tobler et al., 2021) for the 100 bootstrap runs. Figure S8
depicts the log-probability density function used to estimate factors uncertainties. These uncertainties are 2.2 %, 2.2 %, 0.8 %,
1.8 %, 1.6 %, 1.3 %, 0.3 %, and 1.1 % for Shipping 1, Shipping 2, Shipping 3, HOA, COA, OxHOA, MOOA (More Oxidized
Organic Aerosol), and LOOA (Less Oxidized Organic Aerosol) factors, respectively.

## 3.4 Interpretation of PMF solutions

### 3.4.1 Shipping factors

The selected PMF solutions are shown in terms of mass spectra (Figures 5 and 6) and time evolution (Figure 7).

The mass spectra of the three shipping mass spectra are all dominated by hydrocarbon fragments (Figure 5), such as $C_3H_5^+$
(*m/z* 41), $C_3H_7^+$ (*m/z* 43), $C_4H_7^+$ (*m/z* 55), $C_4H_9^+$ (*m/z* 57), $C_5H_9^+$ (*m/z* 69), $C_5H_{11}^+$ (*m/z* 71), along with ions at *m/z* 81
($C_6H_9^+$), 95 ($C_7H_{11}^+$), 97 ($C_7H_{13}^+$), and 111 ($C_8H_{15}^+$), $C_nH_{2n-1}^+$ typical of unsaturated aliphatic compounds, $C_nH_{2n+1}^+$ linked
to saturated alkyl compounds, and $C_nH_{2n-3}^+$ linked to bicycloalkanes and alkynes. These mass spectra are typical of various
combustion sources emissions (McLafferty and Tureček, 1993) including shipping emissions (Fossum et al., 2024; Sun et al.,
2023). The O:C ratios of shipping factors are very low varying from 0.01 to 0.05, H:C ratios from 1.8 to 2.1 (Table S5).



Shipping factors 1 and 2 also present an important contribution of PAHs, with fragments at $m/z$ 128 ($C_{10}H_8^+$) associated with naphthalene, 141 ($C_{11}H_9^+$), 155 ($C_{12}H_{11}^+$), 165 ($C_{13}H_9^+$), 178 ($C_{14}H_{10}^+$) corresponding to anthracene or its isomer phenan-
395 threne, predominant in ship emissions, 179 ($C_{14}H_{11}^+$), 202 ($C_{16}H_{10}^+$) associated with pyrene and its isomers fluoranthene and acephenanthrylene, 205 ($C_{16}H_{13}^+$) and 219 ($C_{17}H_{15}^+$). In agreement with Anders et al. (2023, 2024),we observe PAHs mass spectra dominated by a signal series in $m/z$ sequences of 14 Da corresponding to the addition of a $CH_2$ moiety. In our observations the sequence starts for the alkylated naphthalene at $m/z$ 141-142, and 155-156, as well as for fluorene at $m/z$ 165-166 and 179-180, reflecting the typical PAHs alkylation of ship emissions while in Anders et al. (2023, 2024) and Sippula et al. (2014),
phenanthrene alkylated compounds were predominant. The combustion temperature can both explain the number of rings as well as the degree of substitution, e.g. alkylation (Frenklach, 2002), and it has been shown as large diesel engines show higher alkylation degrees resulting from higher amounts of unburnt fuel Sippula et al. (2014).

The Shipping factor 3 profile exhibits a distinct mass spectrum with relative higher contribution of ion fragments associated to unsaturated aliphatic compounds ($m/z$ 43, 55, 71) and very low PAHs levels. As shown in Figure S9, the SRP between
405 Shipping 3 and Shipping 1 mass spectra reveals major differences between the two spectra, with the predominence of a 14 Da sequence linked to the addition of a $CH_2$ group on the Shipping 3 factor (with ions at $m/z$ 57-58, 71-72, 85-86, 99-100, 113, 127 and 141). Correlation between mass spectra profiles and a HR-TOF-AMS mass spectral datable (MARMOT v3.5A (Jeon et al., 2023; Ulbrich et al., 2009)), are presented in Tables S6 to S13. As expected, the profiles of ship factors show strong correlations with mass spectra from combustion sources as HOA factors and similarity with dioctyl sebacate, known as an
410 additive for engine oils (Kamal et al., 2023; Yu et al., 2021; Shah et al., 2018; Elser et al., 2016; Hu et al., 2016; Crippa et al., 2013; Mohr et al., 2012; Docherty et al., 2011; Aiken et al., 2009), but no information was found on the actual use of these oil in ship fuels.

Table S14 indicates that Shipping factors 1 and 2 present some temporal correlation with $SO_2$ (0.26-0.34), CO (0.12-9.27) BC (0.37-0.56) and $NO_X$ (0.54-0.73) and very good correlation with particles number (0.59-0.78) in the nucleation and Aitken
modes (15 to 70 nm). While shipping factor 3 shows some correlation with CO (0.43), BC (0.4), $NO_X$ (0.52), particles in the accumulation mode (100 to 200 nm) and is only identified for south-westerly winds at higher speeds (above 6 m/s), suggesting that these plumes come from ships at the cruise terminal, as supported by the Non-parametric Wind Regression analysis (NWR) in Figure S10. The low levels of PAHs, the lack of correlation with $SO_2$ and nucleating particles for shipping factor 3 suggest that these ships are equipped with after-treatment devices as scrubbers, in line with Kuittinen et al. (2024) that reported a
decrease in particle-bound PAHs and higher levels of BC, PM and PN above 23 nm compared with combustion of low-sulfur fuels.

The diurnal trends of the shipping factors are presented in Figure S11 and show intense peaks between 7 and 10 a.m. and between 6 and 8 p.m., corresponding to ferry arrivals and departures. It is interesting to note that the baseline of these factors is almost zero when there are no shipping events, which is not the case for the other factors, highlighting the good deconvolution
of the shipping sources (Figure 7).



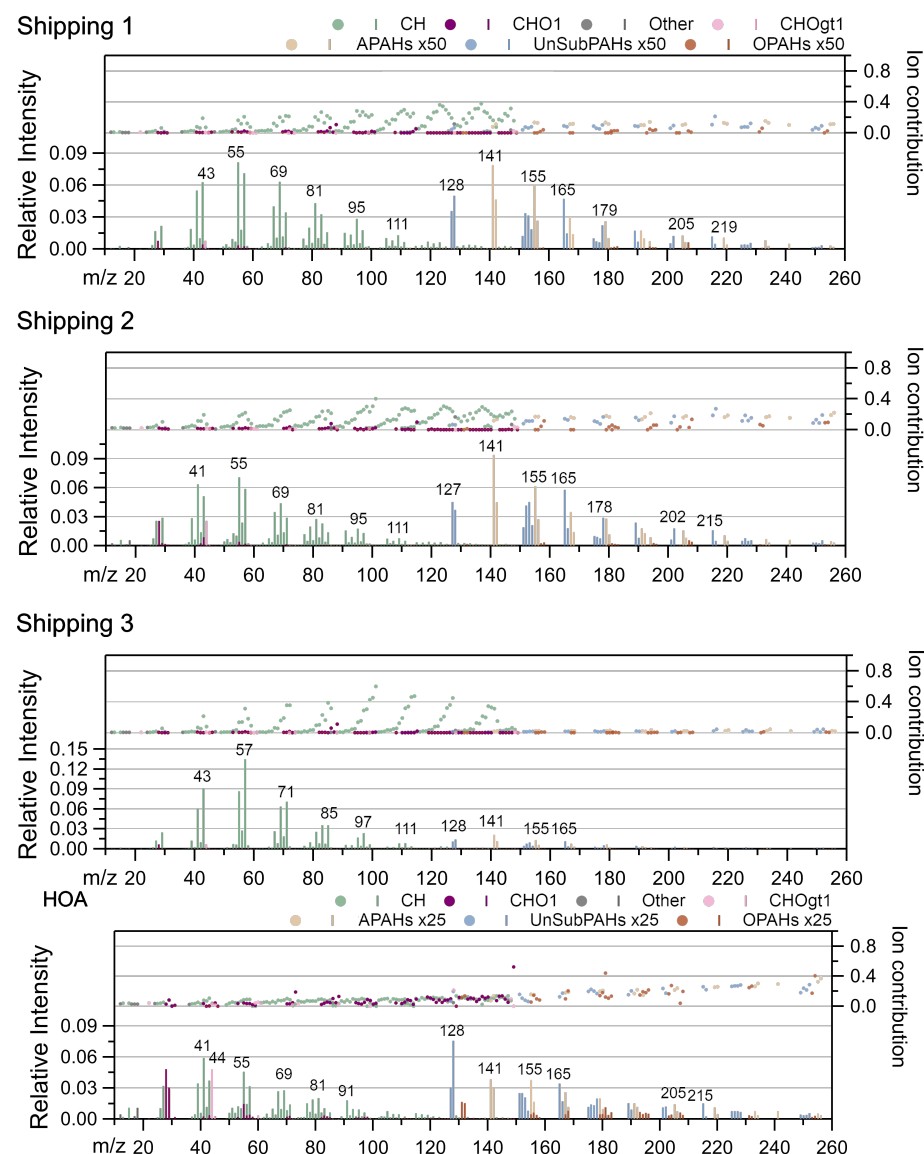

**Figure 5.** Mass spectra of Shipping 1, 2, 3 and HOA factors. PAHs has been multiplied by 50 for all factors except HOA, which PAHs was multiplied by 25. Factor contribution to total contribution is represented by family-colored dots for each ion, with unit CE for each factor.



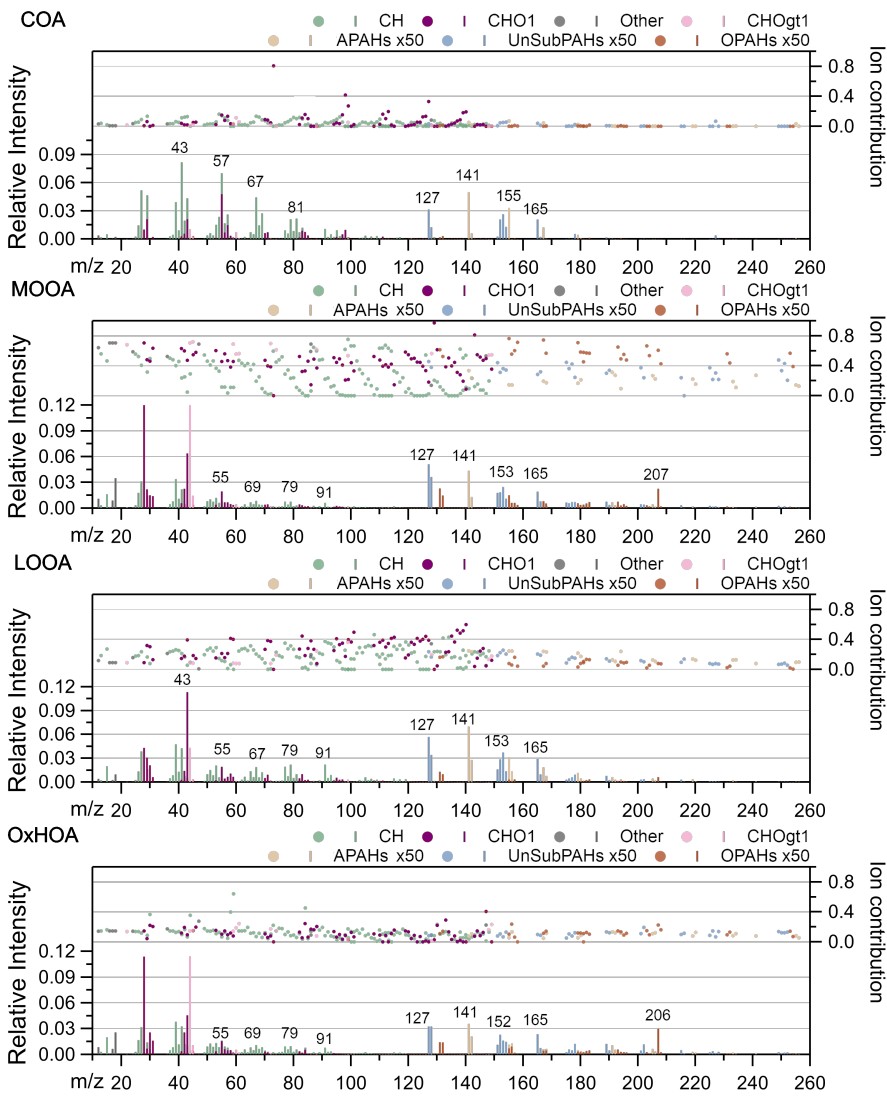

**Figure 6.** Mass spectra of COA, OxHOA, MOOA and LOOA factors. PAHs has been multiplied by 50 for all factors. Factor contribution to total contribution is represented by family-colored dots for each ion, with unit CE for each factor.





**Figure 7.** Time evolution of the individual PMF factors.




### 3.4.2 Hydrocarbon-like Organic Aerosol (HOA)

The predominant ions in the HOA mass spectrum are hydrocarbon fragments (Figure 5) at $m/z$ with $m/z$ 39 ($C_3H_3^+$), 41 ($C_3H_5^+$), 57 ($C_4H_9^+$), 67 ($C_5H_7^+$), 69 ($C_5H_9^+$), 79 ($C_6H_7^+$) and 81 ($C_6H_9^+$), along with oxygenated ions at $m/z$ 28 ($CO^+$), 29 ($CHO^+$), and 55 ($C_3H_3O^+$), all associated to combustion processes (Marimuthu et al., 2020; Goodings et al., 1979). The calculated O:C ratio is 0.12, while shipping factors present O:C ratios below 0.05. The PAHs distribution is quite similar that observed for shipping factors 1 and 2, with highest relative intensities at $m/z$ 128 ($C_{10}H_8^+$) followed by 141 ($C_{11}H_9^+$), 142 ($C_{11}H_{10}^+$), 155 ($C_{12}H_{11}^+$), 165 ($C_{13}H_9^+$), 205 ($C_{16}H_{13}^+$), and 215 ($C_{17}H_{11}^+$). $^+$). These PAHs have been previously reported in road transport emissions (Kostenidou et al., 2021; Muñoz et al., 2018; de Souza and Corrêa, 2016). The HOA mass spectrum contains also an important fraction of oxygenated ions as at $m/z$ 149 ($C_{10}H_{13}O^+$) as well as oxygenated PAHs ($m/z$ 180, 182, 254, Table S4) tentatively assigned to fluorenone, benzopyran and benzo[cd]pyrenone, following previous reports from vehicles emissions by HR-ToF-AMS (Kostenidou et al., 2021). These OPAHs have been associated to after-treatment devices as oxidation catalysts of modern engines (Moldanová, 2025; Kostenidou et al., 2021). The mass spectra discrimination between HOA and shipping factors is illustrated in Figure S12. Spectral Relative Predominance (SRP) highlights similarities and differences across these factors. Major differences are related to oxygenated ions fragments ($C_xH_yO_z^+$), predominant in HOA, such as at $m/z$ 29 ($CHO^+$), 99 ($C_6H_{11}O^+$), 110 ($C_7H_{10}O^+$), 112 ($C_7H_{12}O^+$), 120 ($C_8H_8O^+$), 122 ($C_8H_{10}O^+$), 146 ($C_{10}H_{10}O^+$), 149 ($C_{10}H_{13}O^+$) and also to the OPAHs while shipping factors contain some specific and intense aliphatic fragments as $m/z$ 72, 86, 100, 114, 128 and 140.

As expected, HOA shows good correlation with combustion tracers as CO (0.51), NO$_X$ (0.28) and BC (0.26) and a with PN in the diameter range between 70 and 200 nm (0.34-0.45), Table S14, in accordance with the literature (Chazeau et al., 2022; Marques et al., 2022; Kostenidou et al., 2021; Jaikumar et al., 2017). The HOA mass spectrum correlates well with other published HOA as well as with a COA factor, the theta angle varies between 15 and 23 degrees (Hayes et al., 2013; Hu et al., 2013; Saarikoski et al., 2012), as well as with a source of coal combustion (Table S9). The HOA mass loading (Figure 7) shows maxima in the range 2-3 $\mu g/m^3$ while shipping factors display maxima often exceeding 10 $\mu g/m^3$. Furthermore, the HOA factor is observed under all wind directions (Figure S10) with highest contribution in the proximity of the measurement site which was surrounded by busy roads while a minor contribution is associated with air masses coming from the west and southwest for wind speed exceeding 4 m/s, possibly due to roads behind the cruise terminal. The diurnal profile shows a maximum between 7 and 10 a.m. then it increases again around 6 p.m., peaking at 10 p.m., and slowly decreases until 5 a.m. while shipping factors are characterized by sharp maxima in the morning and the evening but are not observed during nighttime. As HOA maxima (rush hours) occur at the same hours of ship arrivals and departures (Figure S11), some mixing between HOA and shipping factors can occur for air masses coming from west and south-west directions (Figure S10).

### 3.4.3 Cooking-like Organic Aerosol COA

The major ions in the COA mass spectrum are hydrocarbon fragments at $m/z$ 41 ($C_3H_5^+$), 67 ($C_5H_7^+$), 69 ($C_5H_9$)$^+$, 79 ($C_6H_7^+$), and 81 ($C_6H_9^+$), along with oxygenated ions at masses 29 ($CHO^+$), 43 ($C_2H_3O^+$), and 57 ($C_3H_5O^+$) (Figure 6). The COA mass spectrum exhibits a higher contribution compared to other factors of oxygenated ions at $m/z$ 73 ($C_4H_9O^+$), and contributes





around 40 % to ions with $m/z$ 98 ($C_6H_{10}O^+$) and 127 ($C_8H_{15}O^+$). The COA factor exhibited a $m/z$ 55 to $m/z$ 57 ratio of 2.9
and a $m/z$ 67 to $m/z$ 69 ratio of 1.6, aligning with the 2.3–4.5 and 1.1–1.6 ranges reported for aerosol from cooking enriched in
polyunsaturated fatty acids (Pikmann et al., 2024; Xu et al., 2020; Mohr et al., 2012).Conversely, the HOA factor showed a $m/z$
55 to $m/z$ 57 ratio of 1.67 and a $m/z$ 67 to $m/z$ 69 ratio of 0.96, near the 0.63 ± 0.30 average for HOA (Pikmann et al., 2024).
These distinct ratios indicate that COA and HOA have been well separated by the PMF analysis. The COA factor has an O:C
ratio of 0.12 and well correlated with reference COA and oleic acid mass spectra (Table S8), with theta angle between 8 and
465 16 degrees, respectively (Hu et al., 2018; Shah et al., 2018; Elser et al., 2016; Struckmeier et al., 2016; Crippa et al., 2013).
The PAHs contribution to this factor is low, nonetheless, some signals at $m/z$ 127 ($C_{10}H_7^+$) and 128 ($C_{10}H_8^+$) corresponding to
naphthalene, $m/z$ 141 ($C_{11}H_9^+$) and 142 ($C_{11}H_{10}^+$) associated to methyl-naphthalene, $m/z$ 151 ($C_{12}H_7^+$) and 152 ($C_{12}H_8^+$) for
acenaphtylene, and fluorene at $m/z$ 165 ($C_{13}H_9^+$) are observed and are previously reported in COA factors (Cash et al., 2021;
Singh et al., 2016).

The COA factor has a local origin and it is observed for low wind speeds (Figure S10). Its diurnal profile is quite flat during
the day and shows a maximum around 9 p.m. that decreases until 3 a.m. The low values during the day could be explained by
the intense photochemical activity in the region in September. It also shows good temporal correlations with CO (0.39) and PN
between 100 and 200 nm (0.35) (Table S14).

### 3.4.4 Secondary Organic Aerosol (MOOA and LOOA)

Secondary organic aerosol (SOA) factors are characterized by a high fraction of oxygenated ion fragments and are often
differentiate using the relative intensity of ions at $m/z$ 44 ($CO_2^+$) and 43 ($C_2H_3O^+$). The factor with the highest relative intensity
for $m/z$ 44 is defined as MOOA (More-Oxidized OA), while the one with the highest intensity for $m/z$ 43 is defined as LOOA
(Less-Oxidized OA). The MOOA and LOOA factors present an O:C ratio of 0.52 and 0.23, respectively. The apparent low O:C
ratios for these two factors can be explained by the contribution of hydrocarbon ions above $m/z$ 120. Indeed, when considering
an upper limit at $m/z$ 120, as in many previous PMF studies (Elser et al., 2016; Struckmeier et al., 2016; Saarikoski et al., 2012;
Mohr et al., 2012; Docherty et al., 2011), the O:C ratios for MOOA and LOOA become 0.72 for and 0.33 respectively (Table
S14), in agreement with the literature.

The MOOA mass spectrum in Figure 6, is characterized by intense, at $m/z$ 44 ($CO_2^+$) and 28 ($CO^+$), and OPAHs. While
the LOOA mass spectrum has the highest intensity at $m/z$ 43 ($C_2H_3O^+$) and a significant contribution from oxygenated ions
as highlighted by the SRP comparison in Figure S13a. The high contribution of $m/z$ 43 has been tentatively explained by the
sunny Mediterranean summer climate and photochemical activity forming oxygenated OA Struckmeier et al. (2016).

The MOOA factor presents a major contribution from air masses coming from the sea and a minor contribution from local
component. Given that shipping factors did not contain significant levels of OPAHs, their origin could be rather linked to
regional transport of air masses coming from the sea or the Bay area for wind speeds exceeding 2 m/s. In agreement with a
490 previous study in Marseille of Chazeau et al. (2022), the LOOA factor exhibits a strong correlation with nitrate (0.79), even
though this latter is a very minor component of the $PM_1$, while MOOA shows good correlation with time evolution of $SO_4^{2-}$





### 3.4.5 Oxygenated Hydrocarbon-like Organic Aerosol (OxHOA)

The OxHOA mass spectrum (Figure 6) is characterized by intense oxygenated fragments at $m/z$ 28 ($CO^+$), 44 ($CO_2^+$) and minor contribution of ion fragments from hydrocarbons as $m/z$ 27 ($C_2H_3^+$), 39 ($C_3H_3^+$) and 41 ($C_3H_5^+$). Its mass spectrum correlates well with oxygenated factors as MOOA or LOOA from the AMS database (Hu et al., 2016; Crippa et al., 2013; Setyan et al., 2012) (Table S10) but its O:C ratio is of 0.34 placing it between the oxidation degree of primary sources and the secondary factors (Table S5). This can be observed in Figure S14 that depict Ng et al. (2010) triangle for all observed factors. This triangle indicates the area where ambient organic aerosol components typically fall considering the fractions of ions 43 and 44. The MOOA and LOOA clearly fall in this triangle, while OxHOA lies just on the left side of it, showing an intermediate oxidation degree. And the other primary sources (shipping, HOA and COA) occupy the bottom-left corner. The mass spectra of OxHOA and MOOA factors are very similar (cosine similarity of 0.98, Table S16). The SRP in Figure S13b indicates that PAHs and hydrocarbon fragments are enhanced in the OxHOA factor while the oxygenated ion fragments are generally more important in the MOOA factor. The SRP of HOA and OxHOA is presented in Figure S13c and shows as the primary factor, HOA, is enhanced in hydrocarbon ion fragments and PAHs while OxHOA is enriched in small oxygenated fragments as $m/z$ 28 ($CO^+$) and 44 ($CO_2^+$).

The temporal evolution of the OxHOA factor is highly correlated with that of the HOA factor, with a Pearson R value of 0.96 (Table S15). The OxHOA factor also shows quite good temporal correlations with CO (0.64), $NO_X$ (0.37), $NO_3^-$ (0.7), and PN between 100 and 200 nm (0.5) (Table S14). The wind rose of this factor a local character similarly to HOA and COA factors (Figure S10) and has more pronounced contribution from the northwest and south directions (see Figure S15). Finally, the OxHOA factor displays an intermediate level of oxidation between primary and secondary sources and shows similarities with combustion sources and a good correlation with the diurnal pattern of the HOA factor, underscoring its local origin.

### 3.4.6 PAHs Contribution

The Figure 8 offers an overview of the measured OA factors. The three shipping factors, the HOA, the COA and the OxHOA factors represent 11.2 %, 5.6 %, 5.9 % and 12.4 % of the OA mass, respectively. The overall fraction of OA related to transport (road and maritime) is quite high accounting for almost one third of the total OA (29.2 %). The secondary fraction of OA is also very important accounting for 78 % when including the OxHOA factor. High resolution analysis of the HR-ToF-AMS data allowed the identification of PAHs, accounting for 54% for LOOA and MOOA (18 % and 36 %, respectively), 19 % for shipping factors (7 % for Shipping 1, 10 % for Shipping 2 and 1 % for Shipping 3), followed by HOA (15 %), OxHOA (10 %), and COA (2 %).

Altogether, combustion sources account for 51.9 % of the APAHs (shipping factors 28 %, HOA 15.3 % and OxHOA 8.6 %), the other 48 % is distributed among LOOA and MOOA factors underlining the importance of anthropogenic emissions also in the aged aerosol. The OA factors related to transport accounts for 43.6 % of the UnSubPAHs, mostly associated to





naphthalene, acenaphtylene and fluorene. Cooking factor, LOOA and MOOA account for 2.6 % 18.3 % and 35.5 % of the remaining UnSubPAHs. Finally, the OPAHs are more abundant on the MOOA factor (∼60 %), followed by HOA and OxHOA (∼13 %), LOOA (10 %) and shipping factors (3.4 %) suggesting that OPAHs are mostly formed during photo-oxidative processes in the atmosphere. Globally, PAHs accounts for 12.1 ‰ (5.9 ‰ for UnSubPAHs, 3.9 ‰ for APAHs and 2.3 ‰ for OPAHs), while hydrocarbons accounts for 54.5 % and oxygenated compounds for 43.3 % of the total OA measured in Toulon.

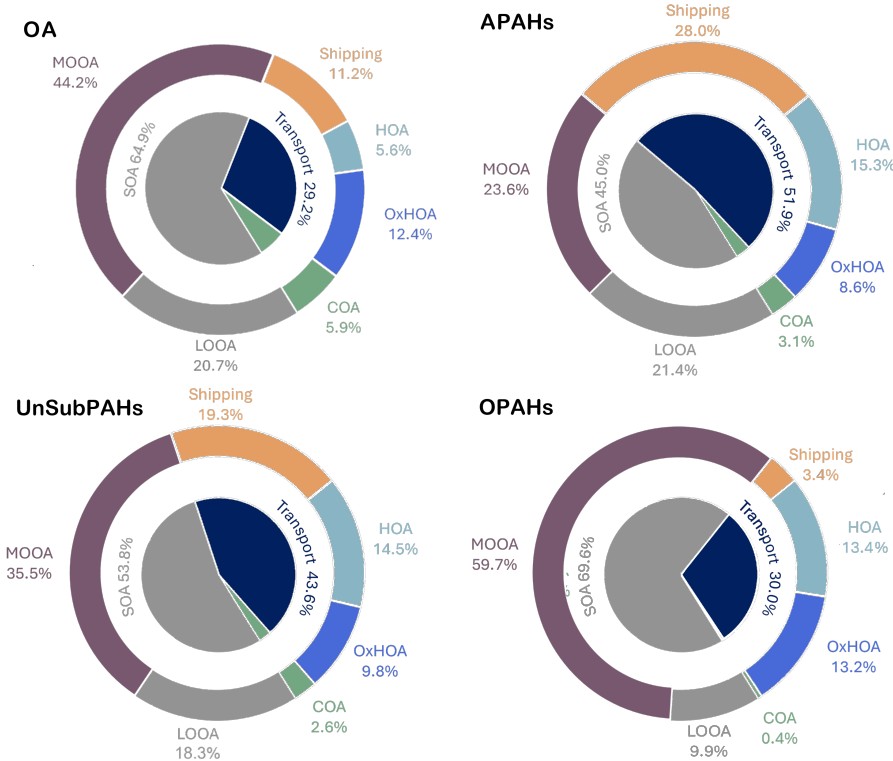

**Figure 8.** Pie chart of OA and PAHs contributions, for the three PAHs families APAHs, UnSubPAHs and OPAHs, across the PMF factors.

## 3.5 Particle number size distribution of PMF factors

The particle number size distribution (PNSD) associated to each factor have been investigated by selecting the ten most intense peaks of each factor. Only data-points having at least ten-minute separation from each other were considered representative of distinct events. The average of the PNSD and its standard deviation for each factor are shown in Figure 9. The Pie charts, corresponding to the average PM$_1$ composition (OA, inorganic ions and BC) associated to each PNSD, are depicted (Figure 9). Ship emissions are known to contribute significantly to ultrafine particle number concentrations. Studies have shown that the combustion of marine fuels, especially in large ships, leads to high emissions of fine particulate matter and UFP (Fossum



et al., 2024; Grigoriadis et al., 2024; Le Berre et al., 2024; Aakko-Saksa et al., 2023; Karjalainen et al., 2022; Anderson et al., 2015).

Our results align with the literature as can be seen in Figure 9 and evidence two typical distributions associated to the identified shipping factors. The PNSD 1 and 2 are associated to UFP with modes around 40-50 nm and are highly associated with Shipping 1 and Shipping 2 factors. The PNSD 1 is explained by shipping factor 1, 2 and 3 (37, 20 and 8 %), BC (21%) and sulfate (6%), highlighting a low impact of sulfur to shipping contributions. The PNSD 2 is associated to shipping factor 1 and 2 (26 and 33%, respectively), BC (14 %) and sulfate (10 %). The very similar size distribution and composition of these two PNSD may suggest that shipping 1 and 2 represent two combustion modes from the same ship rather than emissions from two different vessels, as also suggested by the fact that their emissions are synchronized. PNSD number 3 presents a bimodal distribution at 25 nm and 91 nm and can be explained by Shipping 3 (65%), shipping factors 1 and 2 (13 %), BC (6%) and $SO_4^{2-}$ (5%), the remaining chemical components account for 11% of the PNSD mass. Bimodal distribution presenting a nucleating mode around 15-20 nm and an Aitken mode around 60-100 nm have been reported for vessels running on HFO equipped with exhaust cleaning systems or scrubbers (Fischer et al., 2024; Kuittinen et al., 2024, 2021). Kuittinen et al. (2024) highlight that the use of scrubbers effectively decreased PN below 50 nm and PAHs concentrations while larger particles, typically comprising black carbon (BC) were not affected. Considering PNSD number 3, its chemical composition and wind analyses (Figure S10) it is reasonable to attribute this factor to emissions from vessels equipped with scrubbers. PNSD 1 and 2 are just 1 you do not have two size distributions this is artificial and any reviewer will point it out and ask you for revision. While shipping 1 and shipping 2 have some minor spectral differences and at first sight the ration org/PAHs is different in shipping 1 and 2.

The PNSD number 4 is associated to COA and number 5 to HOA factors, respectively. These distributions are in agreement with the literature reports, with modes at 71 nm and 64 nm (Nursanto et al., 2023; Sowlat et al., 2016) respectively but they overlap each other. PNSD of these factors are definitely impacted by other chemical components as can be seen in Figure 9. PNSD number 4 can be explained by the COA (23%), OxHOA and $SO_4^{2-}$(15%), BC (12%) and shipping 2 (10%). The PNSD number 5 is related to HOA (25%), OxHOA (17%), BC (16%), $SO_4^{2-}$(13%) and LOOA (12%). It is interesting to note that the PNSD 5, linked to HOA emissions, is not affected by COA contribution, underlining the good separation between these sources achieved by the PMF. The PNSD number 6 shows a mode at 82 nm slightly larger than the one of HOA. This distribution is associated with LOOA (27%), $SO_4^{2-}$ (18%), BC (21%), MOOA (12%) and $NO_3^-$(8%). The PNSD number 7 is the largest measured with a mode around 200 nm and it is related to oxidized species as $SO_4^{2-}$ (49%), MOOA (29%), $NO_3^-$ (15%) and BC (12%). The PNSD number 8 is bimodal with a dominant mode around 60 nm and a shoulder around 233 nm. This size distribution is explained by $SO_4^{2-}$ (38%), OxHOA (19%), $NO_3^-$ (15%) and BC (12%).





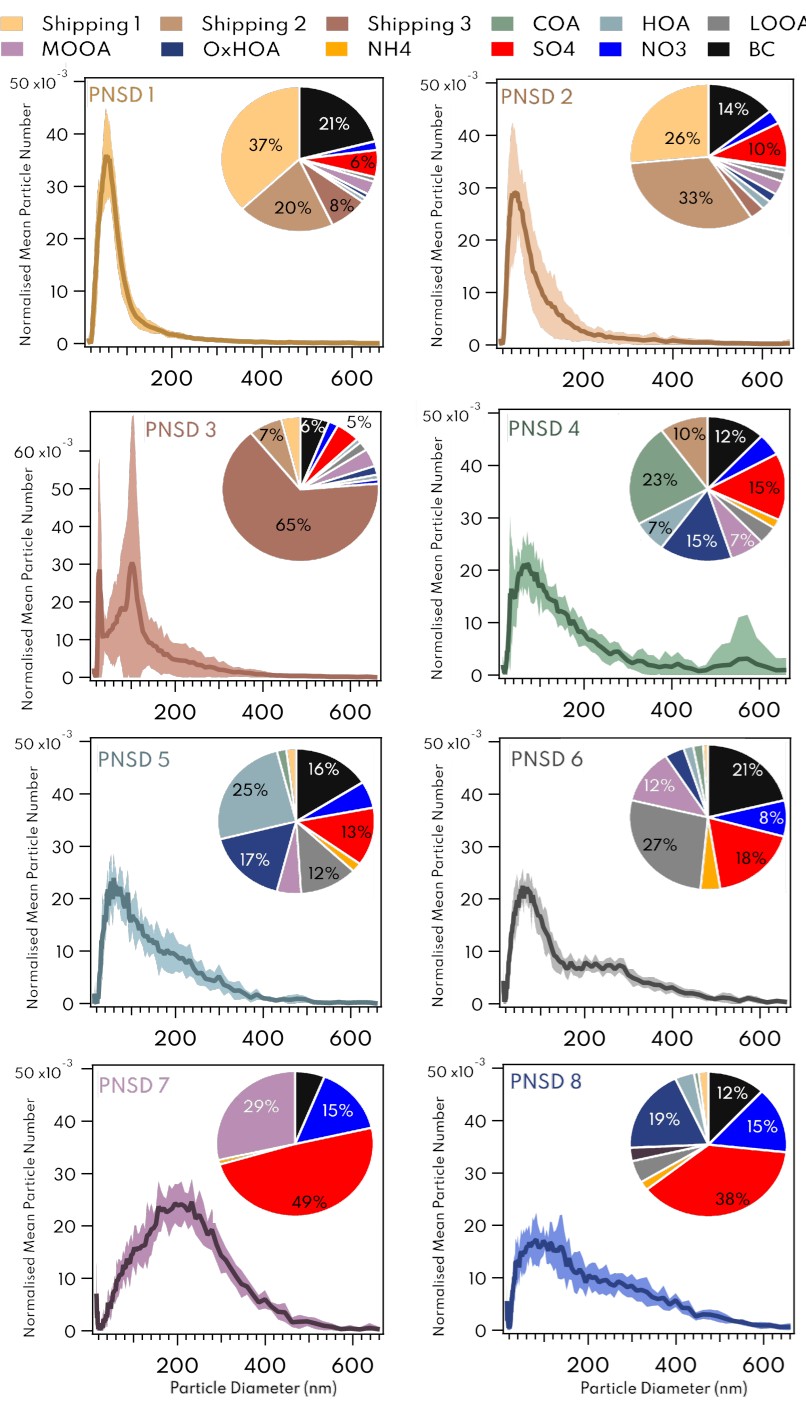

**Figure 9.** PNSD associated to PMF factors, sampled by SMPS, with standard deviation in shaded background. Pie charts represent mass contribution of PMF factors (OA), inorganic species and BC during the 10 most-contributive time-points of each factor.





## 4 Summary and Conclusions

This study, conducted in 2021 in Toulon, a port city on the French Mediterranean coast, assesses emissions from shipping one year after implementing IMO2020 sulfur regulations. The EFs of regulated and non-regulated pollutants have been determined for different types of vessels and, when possible, for various operational phases. The observed reductions in emission factors of sulfur related pollutants, such as $SO_2$ (0.45 g/kg$_{fuel}$), PM sulfates (0.13 g/kg$_{fuel}$), and $NO_X$ (20.7 g/kg$_{fuel}$) reflect the success of successive regulations in mitigating key pollutants, aligning with global efforts to improve air quality in coastal regions. However, levels of BC (0.38 g/kg$_{fuel}$), organics (1.73 g/kg$_{fuel}$), and PAHs (6 mg/kg$_{fuel}$), similar to pre-IMO regulations, underscore the limitations of current regulatory frameworks in addressing the full spectrum of shipping-related pollutants.

Additional PMF analysis of the OA sub-micrometer aerosol fraction was resolved in an 8-factor solution able to separate five primary sources and three aged factors. Overall shipping sources accounted for 11.2 % of the OA (3.7 % from Shipping 1, 4.6 % from Shipping 2, 2.9 % from Shipping 3), the other primary sources are COA 5.9 % and HOA 5.6 %. A partially oxidized combustion source, called OxHOA, could explain up to 12.4 % of the OA. And finally, two secondary sources, LOOA and MOOA explained 20.7 % and 44.2 % of the OA, respectively. The transport sector accounted for nearly 30 % of organic aerosol (OA) mass when considering both the primary sources and the OxHOA factor. The transport sector, considering maritime and road transport together, accounts for almost 52% of the APAHs, 43.6 % of UnSubPAHs, and 30 % of OPAHs. The three shipping factors represent 28 % of total PAHs and are the largest contributor to APAHs at 28 % and a good contributor to UnSubPAHs with 19.3 %, but they represent only 3.5% of the OPAHs. HOA and OxHOA factors show similar contributions to PAHs: 15.3 % and 14.5 % for APAHs, 14.5 % and 9.8 % for UnSubPAHs, and 13.4 % and 13.2 % for OPAHs, respectively.

PNSD analysis highlights how shipping activities represent a major source of UFP in the city of Toulon. Shipping emissions presented either monomodal distribution centered around 50 nm or bimodal distribution (at 25 and 91 nm) typical of vessels equipped with scrubbers. Given the substantial number of UFP emitted by ships, with a mean PN EF of $4.8 \times 10^{15}$ part/kg$_{fuel}$, their high content of PAHs and their ability to penetrate deep into the human respiratory system due to their reduced size, these findings highlight the potential health risks associated with the maritime activities, particularly in densely populated port cities like Toulon. These results also emphasize the importance of advanced source apportionment methods, which enable to differentiate road transport and shipping emissions, thereby improving our understanding of their respective contributions to air quality. Additionally, they provide valuable insights for monitoring emissions in the Mediterranean, particularly in light of the upcoming implementation of a Mediterranean SECA in 2025. This ECA will play a crucial role in improving air quality and reducing maritime pollution in the region as regulatory frameworks continue to evolve.

These findings are critical for shaping future air quality policies, especially as ECA regulations will come into force in the Mediterranean in 2025. This paper highlights the evolving impact of shipping emissions on air quality in a Mediterranean port city following the implementation of IMO2020 sulfur regulations.



*Author contributions.* BD'A conducted the field measurements with the support of BT-R and IX-R. QG performed the analysis and wrote the paper. BD'A and AA designed the research and assured the financial support for the field campaign and the PhD scholarship. All the
authors reviewed and commented the paper.

*Data availability.* Data from the study are available at https://doi.org/10.7910/DVN/S9KF6K (Gunti et al., 2025). More details are available upon request to the corresponding authors.

*Competing interests.* The authors declare they have no conflict of interest.

*Acknowledgements.* We thank Lise Le Berre for sharing her emission factor calculation tool. We are also grateful to Sonia Culi for her
valuable assistance in analyzing ship behavior at the port of Toulon.

*Disclaimer.* The measurement campaign was supported by the projects ANR SHIPAIR (ANR-21-CE22-0015), ADEME PIRATE (2166D0028) and ANRT (n°2022/0244).



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
