# Peer review of "Measurement report: Emission factors and organic aerosol source apportionment of shipping emissions in the coastal city of Toulon, France"

_EGUsphere, 2025_

## Referee Comment (RC1)

**General comments**

The manuscript presents a timely and policy-relevant study that quantifies shipping emission factors and apportions organic aerosol sources in Toulon, France, a coastal port city. The authors report detailed EFs for a wide range of pollutants, including $SO_2$, $NO_x$, CO, $CH_4$, black carbon, organics, PAHs, and particle number concentrations. By distinguishing EFs across vessel types and operational modes (e.g., arrivals vs. departures), and comparing results to pre-regulation studies, the work offers clear evidence of the evolving impact of marine fuel regulations on air quality. The ability to resolve vessel-specific EFs and link them to auxiliary engine power, fuel type, and plume characteristics is a notable strength of the study and represents a valuable dataset for emission inventory improvement and regulatory evaluation. The integration of detailed HR-ToF-AMS measurements with Positive Matrix Factorization factors that distinguish between road- and shipping-related emissions of hydrocarbons represent a valuable and novel contribution to the field by particularly in light of the IMO2020 regulation and the forthcoming SECA Med implementation.

That said, the manuscript would benefit substantially from a comprehensive round of language editing, as many sentences could be rewritten for clarity, conciseness, and smoother flow. The readability of the manuscript would benefit from the consistent use of the Oxford comma, which helps to clarify lists and prevent ambiguity. Additionally, contractions (e.g., "don't" in Line 152) should be avoided in formal scientific writing to maintain a professional tone throughout the text. Section 3.2 would benefit from more direct references to the figures to help guide the reader through the results. Currently, the discussion lacks clear connections to the visual data, which makes it difficult to follow the interpretation and significance of the findings. Overall, the study is technically sound and presents novel results, but significant editorial revision is needed to ensure the presentation matches the quality of the science.

**Specific Comments**

**Lines 186-188.** "The pie chart representing the median PM1 chemical composition shows the following proportions: 53 % organics, 16 % sulfate, 7 % ammonium, 2 % nitrate and 21 % BC."

Could you provide the standard deviations to provide insight of the variability?

**Lines 188-191.** "The most intense PM1 peaks are associated with ship arrivals and departures ..."

Could you provide the quantify the difference in the median and these most intense peaks? How much is this increase in PM1, PN, Org, BC, NOx relative to sulfate? Relative and absolute differences would be good to list here.

**Line 237.** "The $NO_X$ EF from ferries show significant variability, ranging from 5.2 g/kg$_{fuel}$ to 72.5 g/kg$_{fuel}$,"

While it is great to provide the range of values, to best capture the variability, the standard deviation should be provided.

**Line 254.** "though engine design (Tier I for D, Tier II for A, B, C, E)"

Can you explain what the "though engine design" is and what each tier means? This is not explained in the main text nor supplement.

**Lines 275-276.** "This CO EF, ranging from 0.03 g/kg$_{fuel}$ to 1.90 g/kg$_{fuel}$ (median of 0.28 g/kgfuel), show small variation across ship types or vessel size."

Remove "This". "show" should be "shows", but is the implication of small variation across ship types or vessel size based on the listed range? The upper limit of the range is over 6 times the median and the median is over 9 times the lower limit of the range. This tells me that it is highly variable (though, standard deviation should also always be included when discussing variability), so what would support the argument that there is small variation across ship types or vessel size?

**Lines 285-285.** "The highest values are observed for ferries A, B and D characterized by powerful auxiliary engines (more than 6,000 kW cumulated for each ferry)."

Is it possible to correlate auxiliary engine power with Org EF?

**Line 516-517.** "The overall fraction of OA related to transport (road and maritime) is quite high accounting for almost one third of the total OA (29.2 %)."

"Quite high" relative to what? Any reference?

**Section 2.** Can you define the criteria for a plume? How long are each of the plumes? How well does linear interpolation of EFs represent the background? Can you provide any metrics that capture the uncertainty? Perhaps a figure of a plume event which captures the duration of the plume and the variability of EFs would suffice.

**Section 3.2.** A literature table in the supplement with all the references cited in the main text for each EF would be helpful.

**Section 3.2.1.** If more powerful auxiliary engines lead to more $SO_2$ emissions, can you provide a correlation of engine power to $SO_2$ EF? How much of an increase in engine power (kW) results in an increase in EF of $SO_2$?

**Section 3.4.2.** Could you provide an explanation for the diurnal cycle of the cooking organic aerosol lacking meal time peaks in mass? Could this factor rather just have a continental origin rather than cooking?

**Section 3.5.** How does the PNSD when selecting the 10 most intense peaks of each factor compare to the PNSD when each factor makes up a certain threshold of the mass? Can you validate this method in finding a representative PNSD in other ways?

**Figure 1.** The caption lists chloride as a species that is plotted, but according to the legend, it is missing from the figure. Please either include it in the figure or remove from caption and provide reasoning for its absence.

**Figures 2 and 4.** These figures are not referenced anywhere within the main text, which suggests that either 1. they are not significant enough for the main text and should be in the supplement or 2. more text is needed to incorporate these figures. Since Figure 3 is referenced in Section 3.2.5, I suggest that figures 2 and 4 should be referenced in the other sections in Section 3.

**Table S2, S6, S7, S11, S12, S13.** These tables are not referenced anywhere within the main text.

Has work been done to see if the increase in shipping emissions is affected by the number of ships? This is especially relevant as your results section discusses the need to address fleet evolution.

Are there other meteorological parameters such as relative humidity and temperature that are available? Is there any consideration as to how these might affect measurements?

**Technical Comments**

**Line 27.** Explicitly define UFP as ultrafine particles. $PM_1$ and $PM_{0.1}$ are listed but are not explicitly defined as particulate matter with a diameter of ≤1 micron and 0.1 microns, respectively.

**Line 53-54.** "These technologies enable also compliance of environmental regulations for ships running on Heavy Fuel Oil (HFO) with sulfur content exceeding 0.5 % (Laasma et al., 2022)."

Change to "These technologies also allow for the compliance of environmental regulations for ships running on Heavy Fuel Oil (HFO) with sulfur content exceeding 0.5 % (Laasma et al., 2022)."

**Line 55-57.** "The adoption of low-sulfur fuels in maritime transport significantly reduces exposure to fine and ultrafine particle emissions (Mwase et al., 2020),..."

Change to "The adoption of low-sulfur fuels in maritime transport significantly reduces the emission of and related exposure to fine and ultrafine particles (Mwase et al., 2020),..."

**Line 58.** "...despite these advancements„ the use of these new"

Remove the second comma.

**Line 64.** "...transport contribution of 35.2 %and 4.2..."

Add space after the %, before "and".

**Line 96.** "15 and 661.2 nanometers"

This is the only example where nm is written out explicitly. To be consistent with the rest of the paper, replace with "15 nm and 661.2 nm" as done in Lines 546 and 557.

**Line 98.** " which combines a 3080 Electrostatic Classifiers with a Classifiers with a Differential Mobility Analyzer (DMA 3081), a Condensation Particle Counter (CPC 3775), and a 85Kr neutralizer."

Change to " which combines a 3080 Electrostatic Classifier with a Differential Mobility Analyzer (DMA 3081), a Condensation Particle Counter (CPC 3775), and a 85Kr neutralizer."

**Lines 99-100.** "$CO_2$, CO and $CH_4$ was measured by Picarro analyzer..."

Change to "$CO_2$, CO and $CH_4$ were measured by a Picarro analyzer"

**Line 109.** "m/z ratios"

Change to "mass-to-charge ratios (m/z)".

**Lines 123-124.** "The background has been estimated from measurements taken before and after the plume event."

This is repetitive to lines 119 to 120: "We use linear fit-based EFs that linearly interpolate background concentrations between the levels before and after the plume...". I suggest removing Lines 123-124.

**Line 142.** "...is a method increasingly used to calculate correlation when comparing mass spectra"

Add "the" before "correlation".

**Lines 185-186.** "including meteorological data (wind speed and direction) HR-ToF-AMS species and black carbon, as well as PN, PM and gases ($NO_X$ and $SO_2$)."

This is difficult to read. I am not sure what are the criteria for adding commas. I suggest, "including meteorological data (wind speed and direction), aerosol composition (HR-ToF-AMS species and black carbon), as well as PN, PM, and gases ($NO_X$ and $SO_2$)."

**Line 189.** "130 to 290° (northwest to east)"

If 290° is considered northwest, then 130° should be considered southeast. Lines 205-206 define "130 to 290°" as southeast to northwest, so be consistent.

**Lines 203-205** "When particle number concentration peak exceeds twice the average background level and originates from the sea is attributed to a ship plume."

Change to "When the peak in particle number concentration exceeds twice the average background level and originates from the sea, the source is attributed to a ship plume."

**Line 217.** "below the IMO2020 requirements of 0.5 %, the mean $SO_2$ EF"

Replace the comma after 0.5% with a semicolon.

**Line 245.** "consistent with Celik et al. (2020) (20±3 g/kg$_{fuel}$)"

Change to "consistent with a previously reported value of 20±3 g/kg$_{fuel}$ (Celik et al., 2020)".

**Line 247.** "...four times higher than the median EF reported in Marseille by (Le Berre et al., 2024) (5.4 g/kg$_{fuel}$)..."

Change to **"**higher than the median EF of 5.4 g/kg$_{fuel}$ reported in Marseille by Le Berre et al. (2024)". When directly referring to a study (such as here when using "by"), only the year should be in parenthesis. This was correctly done in Lines 72, 234, and 263, for example.

**Line 249.** "...as reported by (Bai et al., 2020)."

Change to "as reported by Bai et al. (2020).

**Line 250.** "The EF of CO vary from..."

Change "vary" to "varies".

**Lines 256-257.** "The median CH$_4$ EF of 1.0 g/kg$_{fuel}$ is slightly higher than values reported from Marseille in 2021 of median 0.4 g/kg$_{fuel}$ (Le Berre et al., 2024) and similar to reports from Volent et al. (2025) (median of 0.99 g/kg$_{fuel}$)."

Change to "The median CH$_4$ EF of 1.0 g/kg$_{fuel}$ is slightly higher than the median value of 0.4 g/kg$_{fuel}$ reported from Marseille in 2021 (Le Berre et al., 2024) but similar to the median value of 0.99 g/kgfuel reported by Volent et al. (2025)."

**Lines 257-258.** "This EF is considerably higher than other studies with EFs of 0.02 g/kg$_{fuel}$ and 0.05 g/kg$_{fuel}$ from (Cooper, 2003; Timonen et al., 2022),respectively."

Change to "This EF is considerably higher than other studies with EFs of 0.02 g/kg$_{fuel}$ (Cooper et al., 2003) and 0.05 g/kg$_{fuel}$ (Timonen et al., 2022)."

**Lines 259-260.** "Among the ships identified none was LNG-fueled, but the yachts exhibited highest CH$_4$ emissions..."

Change to "Among the identified ships, none were LNG-fueled, but the yachts exhibited the highest CH$_4$ emissions..."

**Lines 261-262.** "Inadequately tuned engines, such as those on yachts tend to emit much more methane, in line with the emissions for small vessels as previously reported by Wang et al. (2022) (5.2 g/kg$_{fuel}$)."

Change to "Inadequately tuned engines, such as those on yachts, tend to emit much more methane, in line with a reported EF of 5.2 g/kg$_{fuel}$ for small vessels (Wang et al., 2022)."

**Lines 273-275.** "The mean CO EF of 0.38 g/kg$_{fuel}$ is consistent with literature reported in Marseille in 2021(Le Berre et al., 2024) (0.48 g/kg$_{fuel}$ for maneuvering ships) and with a cargo vessel (0.48 g/kg$_{fuel}$) (Huang et al., 2018) and reflects typical emissions from diesel marine engines using low-sulfur fuels."

Change to "The mean CO EF of 0.38 g/kg$_{fuel}$ is consistent with previously reported CO EFs of 0.48 g/kg$_{fuel}$ for maneuvering ships in Marseille in 2021 (Le Berre et al., 2024) and for cargo vessels (Huang et al., 2018), reflecting typical emissions from diesel marine engines using low-sulfur fuels."

**Lines 280-281.** "...in quite good agreement with literature values reported by Le Berre et al. (2024) (median 0.86 g/kgfuel),Celik et al. (2020) (mean 3.0 g/kgfuel), and Diesch et al. (2013) (mean 1.8 g/kgfuel)."

Change to "...in quite good agreement to a previously reported median value of 0.86 g/kgfuel (Le Berre et al., 2024) and mean values of 3.0 g/kgfuel (Celik et al., 2020) and 1.8 g/kgfuel (Diesch et al., 2013)."

**Line 286** The EF of PAHs, corresponding to the sum of PAHs families defined by Herring et al. (2015), exhibits a mean value of 10.3 mg/kg$_{fuel}$, aligning closely with values from Celik et al. (2020) (mean 11 mg/kg$_{fuel}$) and Diesch et al. (2013) (mean 5.3 mg/kg$_{fuel}$).

Change to "The EF of PAHs, corresponding to the sum of PAHs families defined by Herring et al. (2015), exhibits a mean value of 10.3 mg/kg$_{fuel}$, aligning closely with reported mean values of 11 mg/kg$_{fuel}$ (Celik et al., 2020) and 5.3 mg/kg$_{fuel}$ (Diesch et al., 2013)."

**Lines 292-294.** "The mean EF for $SO_4^{2-}$ is 0.13 g/kg$_{fuel}$, reflecting a significant decreased compared to previous reports, being approximately 30 times lower than the 4 g/kg$_{fuel}$ value reported by (Celik et al., 2020) and 4 times lower than that of Diesch et al. (2013) (0.54 g/kg$_{fuel}$). This reduction is consistent with more recent findings of Le Berre et al. (2024) (median of 0.05 g/kg$_{fuel}$)."

Change to "The mean EF for $SO_4^{2-}$ is 0.13 g/kg$_{fuel}$, approximately 4-30 times lower than the $SO_4^{2-}$ EF values reported by Diesch et al. (2013) and Celik et al. (2020), but consistent with a more recent study that found a median $SO_4^{2-}$ EF of 0.05 g/kg$_{fuel}$ (Le Berre et al., 2024)."

**Line 461.** Add space before "Conversely".

**Line 588. "...**due to their reduced size,"...

Replace "reduced" with "smaller".

---

## Referee Comment (RC2)

**Overall comments:**

The study presents monitoring of a series of air pollutants, e.g., SO2, NOx, CO, CH4, PM (black carbon, ions, PAHs and other organic matters), near ship terminals in the coastal city of Toulon, France. The dataset enables the derivation of emission factors (EFs) for these pollutants, while the application of positive matrix factorization (PMF) to HR-ToF-AMS measurements of organic aerosols (OA) allows quantification of different emission sources, including the contribution of shipping to local air pollution. The dataset is comprehensive, and the results and discussion are generally sound and logically presented. The findings contribute to methodologies for source apportionment, particularly for assessing shipping emissions in coastal areas, and provide valuable insights for implementing Emission Control Area (ECA) regulations in the Mediterranean region. However, several aspects require clarification and improvement before the manuscript can be considered for publication. I recommend a major revision, with detailed comments and suggestions provided below.

**Detailed comments:**

It is known that combustion-emitted organic compounds such as PAHs can partition between the gas and particle phases. For more volatile PAHs like naphthalene, gaseous concentrations are typically much higher than those in the particle phase. It seems that only particle-bound PAHs were analyzed in this study. What is the approximate fraction of the EF of particle-bound PAHs relative to total PAH EFs (including gaseous PAHs)? Given that gas-particle partitioning is temperature-dependent, to what extent might temperature influence the calculated EFs and the interpretation of the PMF results?

In lines 67-77: The authors mentioned the challenges of using PMF to apportion sources of OA, such as the merging of multiple sources into a single factor and overlapping mass spectral patterns. How does this study address these challenges and fill these gaps? What advanced or novel techniques were applied to resolve these issues? Including such a description somewhere in the manuscript would be beneficial and would enhance the significance of this work from a methodological perspective.

Figure S1 shows that there is a large military port near the study area. Are military vessels incorporated in the analysis of this study? In addition, military bases may be a significant source of OA. How much is known about military emissions, and could they influence the results of this study?

Equation 1: I think the equation should be  $EF = \frac{\int_E^T[x](t)dt}{\int_E^T[CO](t)dt} \times \dots$ , correct me if I was wrong.

In the caption of Figure 1, how were the organic peaks selected? It was stated that they represent the highest peaks (line 337), but this does not seem entirely accurate. For example, peaks 6 and 7 appear lower than some subpeaks around peaks 2 and 9. There are also several other signature peaks visible. Why were only ten peaks selected? It would be worthwhile to clarify this selection criterion.

Line 200-207: How was the contribution of ship emissions critically evaluated when the wind direction was blowing seaward rather than toward the observation site? I would expect that the instruments could not capture the plume in such cases. If so, although these emissions might not significantly affect local populations, they are still released into the atmosphere and could impact elsewhere.

Consider merging figure 3 into Figure 4.

I am curious about how the plumes were allocated to different ship emissions. What is the approximate lag time between plume emission and detection by the instruments? Is this lag time variable under different wind speeds? If so, this variability could affect the accuracy of source allocation among ship types. Additional explanation on this point would be helpful.

It may be worthwhile to include a brief description of the instrumentation in the Supplement, such as the operating parameters for the HR-ToF-AMS and some others, even though the source of the detailed method has been cited in the main text.

Line 179 and line 364-371, "a-value" is not easy to understand. Why was a range of 0 to 0.3 tested for the three shipping constraints, whereas a broader range of 0 to 1 was used for HOA and COA? Additional explanations may be helpful.

**Small things:**

Line 7: add "analysis" after PMF sounds more readable?

Line 24: seems "such" was missing before "as"

Line 58: doubled commas between "advancement" and "the"

Line 221, add "reported by Sinha et al. (2023)" or change to "(Sinha et al., 2023)"

Line 238, "with Zhang et al. (2024)'s findings"

Line 243, "the need of further measurements" sounds more natural.

Line 250-251: suggest rephrasing as "The EF of CO vary from 2.43 g/kgfuel to 57.87 g/kgfuel (median of 20.6 g/kgfuel), reflecting that factors like engine star-.."

Line 258, seems a space was missing before "respectively"

Line 258-259, "Among the ships identified none was LNG-fueled" is not clear, a comma was missing between "identified" and "none"?

Line 273-274; suggest rephrasing as "The mean CO EF of 0.38 g/kgfuel is consistent with reported EFs in Marseille in 2021(Le Berre et al., 2024) (0.48 g/kgfuel for maneuvering ships) and from a cargo vessel (0.48 g/kgfuel) (Huang et al., 2018), reflecting typical emissions from diesel..."

Line 276, "sizes"

Line 280-283: "EFs"

Line 289-290: this sentence is not complete. Should be something like "This pattern underscores the impacts of operation practices (such as...) on PAH EFs.

Line 292, "lower than the value of 4 g/kg reported by .."

Line 305: "Only cruise ships were equipped with scrubbers and were associated with..."?

Line 310: "understand shipping impact on air quality in coastal cities."

Line 340: "measured" rather than "true"

Line 402: "(Sippula et al., 201)" rather than "Sippula et al. (2014)"

Line 451:" peaks" rather than "peaking"

Line 464: "Table S9" rather than "Table S8"?

Line 476: "differentiated" rather than "differentiate"

Line 510-511 "The wind rose of this factor a local character similarly to HOA and COA

factors and ..." The sentence is not clear.

Line 523-524: the sentence is not clear.

Line 528-529, is the global data originated from literature? If so, please include reference.

Line 545: Starting from PNSD 3, "number" was added between "PNSD" and the digit (not the case for PNSD 1 and 2). I would suggest keeping them consistent.

Line 550: "highlighted"

Line 557-558, I did see the point of this sentence. Was it a mistake?

Line 583, was the contribution of shipping factors to total PAHs (28%) accidently the same to the contribution to APAHs (28%)?

---

## Author Comment (AC1)

**General comments**

*The manuscript presents a timely and policy-relevant study that quantifies shipping emission factors and apportions organic aerosol sources in Toulon, France, a coastal port city. The authors report detailed EFs for a wide range of pollutants, including $SO_2$, $NO_x$, CO, $CH_4$, black carbon, organics, PAHs, and particle number concentrations. By distinguishing EFs across vessel types and operational modes (e.g., arrivals vs. departures) and comparing results to pre-regulation studies, the work offers clear evidence of the evolving impact of marine fuel regulations on air quality. The ability to resolve vessel-specific EFs and link them to auxiliary engine power, fuel type, and plume characteristics is a notable strength of the study and represents a valuable dataset for emission inventory improvement and regulatory evaluation. The integration of detailed HR-ToF-AMS measurements with Positive Matrix Factorization factors that distinguish between road- and shipping-related emissions of hydrocarbons represents a valuable and novel contribution to the field, particularly in light of the IMO2020 regulation and the forthcoming SECA Med implementation.*

*That said, the manuscript would benefit substantially from a comprehensive round of language editing, as many sentences could be rewritten for clarity, conciseness, and smoother flow. The readability of the manuscript would benefit from the consistent use of the Oxford comma, which helps to clarify lists and prevent ambiguity. Additionally, contractions (e.g., "don't" in Line 152) should be avoided in formal scientific writing to maintain a professional tone throughout the text. Section 3.2 would benefit from more direct references to the figures to help guide the reader through the results. Currently, the discussion lacks clear connections to the visual data, which makes it difficult to follow the interpretation and significance of the findings. Overall, the study is technically sound and presents novel results, but significant editorial revision is needed to ensure the presentation matches the quality of the science.*

**Response:**

We sincerely thank the reviewer for this very positive and constructive overall evaluation of our work.

In response to these general comments, we conducted a thorough editorial revision of the entire manuscript to enhance clarity, conciseness, and stylistic consistency. Contractions have been removed, the Oxford comma has been applied systematically, and sentence structure has been refined to improve readability and scientific flow.

We also reinforced the connection between the discussion and the visual material, especially in Section 3.2, by explicitly referencing the relevant figures and clarifying their interpretation.

All editorial and stylistic revisions suggested by the reviewer have been implemented consistently throughout the manuscript. These changes are highlighted in blue in the revised version for transparency.
* * *
**Reviewer comment:**
*Lines 186–188. "The pie chart representing the median PM1 chemical composition shows the*

*following proportions: 53 % organics, 16 % sulfate, 7 % ammonium, 2 % nitrate and 21 % BC."*
*Could you provide the standard deviations to provide insight of the variability?*

**Response:**
The standard deviations of each $PM_1$ chemical species have now been calculated and are reported both in the text and in the corresponding figure caption. The values are as follows: organics (±13 %), sulfate (±9 %), ammonium (±4 %), nitrate (±1 %), and BC (±14 %). This addition highlights that organics and BC are the dominant and most variable components of $PM_1$. The pie chart continues to represent the median composition, which better reflects the central tendency of the campaign dataset while minimizing the influence of high-concentration events.
* * *
**Reviewer comment:**
*Lines 188–191. "The most intense $PM_1$ peaks are associated with ship arrivals and departures ..."*
*Could you quantify the difference between the median and these most intense peaks? How much is this increase in $PM_1$, PN, Org, BC, $NO_x$ relative to sulfate? Relative and absolute differences would be good to list here.*

**Response:**

We quantified the difference between (i) the median concentration computed across all identified plumes and (ii) the maximum value reached during the most intense plume for $PM_1$, PN, Org, BC, $NO_x$ and $SO_4^{2-}$. For each species, we report both the absolute increase (max – median) and the relative increase (max / median).

Using the plume medians and the corresponding maximum values, we obtain:

- $PM_1$: increase of +21 $\mu g.m^{-3}$, factor 3.1, max 152 $\mu g.m^{-3}$

- PN: increase of +2.4 × $10^4$ $cm^{-3}$, factor 3.4, max 6.0 × $10^4$ $cm^{-3}$

- Org: increase of +15 $\mu g.m^{-3}$, factor 5.1, max 112 $\mu g.m^{-3}$

- BC: increase of +2.9 $\mu g.m^{-3}$, factor 3.6, max 14 $\mu g.m^{-3}$

- $NO_x$: increase of +123 $\mu g.m^{-3}$ (≈ 65 ppb), factor 3.3, max 470 $\mu g.m^{-3}$

- $SO_4^{2-}$: increase of +0.27 $\mu g.m^{-3}$, factor 1.20, max 6.8 $\mu g.m^{-3}$

To compare these enhancements with sulfate, we normalized each relative increase to the sulfate factor. The resulting ratios are:

- $PM_1$: 2.6 × sulfate

- PN: 2.9 × sulfate

- Org: 4.2 × sulfate

- BC: 3.0 × sulfate

- $NO_x$: 2.8 × sulfate

**Change in the manuscript: (lines 214 - 218)**

*"When comparing the plume median to the maximum concentration reached during the most intense events, $PM_1$ increased by a factor of about 3.1 (≈ +21 $\mu g.m^{-3}$), PN by 3.4 (≈ +2.4 × $10^4$ $cm^{-3}$), Org by 5.1 (≈ +97 $\mu g.m^{-3}$), BC by 3.6 (≈ +11 $\mu g\ m^{-3}$), and $NO_x$ by 3.3 (≈ +123 $\mu g\ m^{-3}$). In contrast, sulfate increased only by a factor of about 1.2 (≈ +0.27 $\mu g\ m^{-3}$), confirming its much weaker variability during peak plumes."*
* * *
**Reviewer comment:**
Line 237. *"The $NO_x$ EF from ferries show significant variability, ranging from 5.2 g/kgfuel to 72.5 g/kgfuel."* While it is great to provide the range of values, to best capture the variability, the standard deviation should be provided.

**Response:**
We have now calculated and included the standard deviations of the emission factors in both the main text and Table 2. Reporting the mean, median, range, and standard deviation for each pollutant provides a more comprehensive representation of the variability observed among different vessels and operational modes.
This information has been explicitly mentioned in all EFs' paragraphs.
* * *
**Reviewer comment:**
Line 254. *"though engine design (Tier I for D, Tier II for A, B, C, E)."* Can you explain what the *"though engine design"* is and what each tier means? This is not explained in the main text nor supplement.

**Response:**
The phrase "though engine design" referred to differences in engine generation and it is related to IMO $NO_x$ emission standards (Tiers I–III) defined under MARPOL Annex VI. These tiers establish maximum allowable $NO_x$ emissions per unit of engine power as a function of rated speed and year of construction. Tier I, applicable to engines built after 2000, allows 17.0 g $kWh^{-1}$ at 130 rpm, decreasing to 9.8 g $kWh^{-1}$ at 2000 rpm. Tier II (after 2011) tightens these limits to 14.4 g $kWh^{-1}$ at 130 rpm and 7.7 g $kWh^{-1}$ at 2000 rpm. Tier III, applicable to engines installed within $NO_x$ Emission Control Areas (NECAs)—a subset of Emission Control Areas (ECAs)—after 2016, further reduces these limits to 3.4 g $kWh^{-1}$ at 130 rpm and 2.0 g $kWh^{-1}$ at 2000 rpm.
In our dataset, ship D is equipped with a Tier I engine, while ships A, B, C, and E are fitted with Tier II engines.

**Change in the manuscript: lines (51 - 53)**

*"In addition to the limits on fuel sulfur content, MARPOL Annex VI also establishes progressive NOx emission standards, known as Tier I, Tier II, and Tier III. These tiers define the maximum allowable NOx emissions per unit of engine power as a function of the engine's rated speed."*
* * *
**Reviewer comment:**
Lines 275–276. *"This CO EF, ranging from 0.03 g/kgfuel to 1.90 g/kgfuel (median of 0.28 g/kgfuel), show small variation across ship types or vessel size."* Remove *"This"*. *"show"* should be

*"shows", but is the implication of small variation across ship types or vessel size based on the listed range? The upper limit of the range is over 6 times the median and the median is over 9 times the lower limit of the range. This tells me that it is highly variable (though, standard deviation should also always be included when discussing variability), so what would support the argument that there is small variation across ship types or vessel size?*

**Response:**
We thank the reviewer for catching the error and the inconsistency in interpretation. The statement indeed referred to BC rather than CO. We agree that the expression "small variation" was misleading.

The sentence has been corrected as follows (lines 307 - 311):

"The BC EFs exhibit a broad range of values (0.03–1.90 $g \cdot kg^{-1}$ fuel), primarily due to a few isolated plumes characterized by unusually high values. These outliers are likely linked to transient conditions such as low engine load or incomplete combustion, rather than systematic differences among ship types or vessel sizes. When excluding these extreme points, the interquartile range (0.09–0.42 $g \cdot kg^{-1}$ fuel) suggests that most of the plumes fall within a relatively consistent range across vessels under typical operating conditions."
* * *
**Reviewer comment:**
*Lines 285–286. "The highest values are observed for ferries A, B and D characterized by powerful auxiliary engines (more than 6,000 kW cumulated for each ferry)." Is it possible to correlate auxiliary engine power with Org EF?*

**Response:**
We calculated the correlations between organic aerosol (Org) emission factors and both auxiliary and main engine power for all maneuvers as well as for arrivals and departures separately (Table S3).

No significant correlation was found between Org EFs and either auxiliary or main engine power when across all maneuvers ($R^2 < 0.15$). During arrivals, a weak negative trend was observed with auxiliary power was observed (slope = $-1.86 \times 10^{-4} \pm 2.16 \times 10^{-4}$ $g\,kg^{-1}$ fuel $kW^{-1}$, $R^2 = 0.20$), whereas during departures, no clear relation emerged ($R^2 \approx 0$). The correlation with main engine power remained low in all cases ($R^2 \leq 0.33$).

These results suggest that organic aerosol emissions are not governed by installed engine power, but rather by operational and ship-specific parameters such as fuel combustion efficiency.

All regression results are summarized in Table S4 and the corresponding plots are presented in Figure S6, which includes $SO_2$ and Org EFs as a function of auxiliary, main, and total engine power for arrivals, departures, and combined maneuvers.

**Change in the manuscript: (lines 319 - 322)**

*"However, no significant correlation was found between Org EFs and either auxiliary, main, or total engine power ($R^2 < 0.20$), indicating that fuel composition, combustion efficiency, and transient operating conditions exert a stronger influence than engine size. Regression statistics*

*for arrivals, departures, and combined maneuvers are summarized in Table S4, and the corresponding relationships are illustrated in Figure S6."*
* * *
**Reviewer comment:**
*Lines 516–517. "The overall fraction of OA related to transport (road and maritime) is quite high accounting for almost one third of the total OA (29.2 %)." "Quite high" relative to what? Any reference?*

**Response:**
The expression "quite high" referred to the fact that the fraction of OA attributed to transport sources (road + maritime) in our PMF analysis (29.2 % of total OA) is near the upper end of values reported for coastal and port environments.
Although few studies directly apportion OA from shipping, comparable receptor-model analyses report lower or similar values. Broader receptor-model studies including both inorganic and organic species generally report total shipping contributions between 5 % and 20 % of $PM_{2.5}$ or $PM_{10}$ (Wu2019, Bove2016, Pandolfi2011, Minguillon2008). Therefore, the 29 % of OA attributed to transport in Toulon falls at the upper end of values reported literature for port or near port environments. These comparison values and references have now been added directly to the manuscript to clarify the context and substantiate the interpretation.

**Addition to the manuscript: (lines 564 - 567)**

*"The overall fraction of OA related to transport (road and maritime) account for 29.2 % of total OA. Broader receptor-model studies including both organic and inorganic species generally estimate 5–20 % of PM2.5 or PM10 from maritime sources (Wu et al., 2019; Bove et al., 2016; Pandolfi et al., 2011; Minguillón et al., 2008)."*
* * *
**Reviewer comment:**
*Section 2. Can you define the criteria for a plume? How long are each of the plumes? How well does linear interpolation of EFs represent the background? Can you provide any metrics that capture the uncertainty? Perhaps a figure of a plume event which captures the duration of the plume and the variability of EFs would suffice.*

**Response:**
In this study, a plume is defined as a transient enhancement in particle number concentration (CPC) exceeding at least twice the local background, with a mean wind direction between 130° and 290° (from southeast to northwest) and a wind direction standard deviation below 30°. The start and end times are determined from concurrent increases in CPC and $CO_2$, adjusted to the instrumental time steps. Typical plume durations range from a few minutes up to about 20 minutes, depending on ship distance and meteorological conditions.

The background concentration is determined by linear interpolation between pre- and post-plume periods, which provides results consistent with other background estimation methods (rolling or median; Volent et al., 2025). Each emission factor (EF) is automatically calculated and then manually validated to ensure that only genuine ship plumes are retained, with clearly

defined start and end times for each event. The plume boundaries are individually adjusted for each pollutant to account for slight desynchronization between instruments, ensuring accurate integration of excess concentrations.

This combined automatic and manual validation ensures that each EF corresponds to a well-defined emission event. A representative plume example showing the background interpolation, plume boundaries, and variability in measured pollutants has been added in Figure S5 in the Supplementary Information.

**Change in the manuscript: (lines 156 - 167)**

*"In this study, a plume was defined as a transient enhancement in particle number concentration (CPC) exceeding at least twice the local background, observed under a mean wind direction between 130° and 290° (from southeast to northwest) and a wind direction standard deviation below 30°. The start and end times were determined based on concurrent increases in PN and $CO_2$ concentration, adjusted to the instrumental time resolution. Plume durations ranged from a few minutes to approximately 20 minutes, depending on the ship's distance from site and the prevailing meteorological conditions.*

*Each emission factor (EF) was automatically calculated using an emission factor calculation tool (Le Berre et al., 2024), based on the carbon mass balance method, and subsequently manually validated to ensure that only genuine ship plumes were retained, with clearly defined start and end times for each event. The plume boundaries were individually adjusted for each pollutant to account for slight desynchronization between instruments, thereby ensuring accurate integration of excess concentrations. This combination of automatic and manual validation ensures that each EF corresponds to a well-defined transient emission event. An illustrative example of a plume, including the background interpolation and pollutant variability, is provided in Figure S5."*

**Reviewer comment:**
*Section 3.2. A literature table in the supplement with all the references cited in the main text for each EF would be helpful.*

**Response:**
A new table (Table S5) has been added to the Supplementary Information, compiling all literature emission factor (EF) values cited in the main text. The table lists, for each pollutant, the reported EF ranges, measurement type (on-board, in-plume, or test-bench), fuel type, engine tier, and corresponding references. This addition provides a comprehensive overview of the datasets used for comparison and later discussed in Section 3.2.
* * *
**Reviewer comment:**

*Section 3.2.1. If more powerful auxiliary engines lead to more $SO_2$ emissions, can you provide a correlation of engine power to $SO_2$ EF? How much of an increase in engine power (kW) results in an increase in EF of $SO_2$?*

**Response:**
We recalculated the correlations between $SO_2$ emission factors (EFs) and both auxiliary and main engine power, considering all maneuvers as well as arrivals and departures separately (Table S3).

Over all maneuvers, a moderate correlation was found between $SO_2$ EF and auxiliary engine power (slope = $1.48 \times 10^{-4} \pm 7.14 \times 10^{-5}$ g kg$^{-1}$ fuel kW$^{-1}$, $R^2 = 0.35$), while the correlation with total installed engine power was negligible ($R^2 = 0.00$).

When separating the operational modes, correlation with auxiliary engine power strengthened during departures (slope = $2.25 \times 10^{-4} \pm 1.05 \times 10^{-4}$, $R^2 = 0.60$) and weakened during arrivals (slope = $7.03 \times 10^{-5} \pm 9.77 \times 10^{-5}$, $R^2 = 0.15$).

Correlations with main engine power were generally weaker ($R^2 < 0.4$).

On average, an increase of 1,000 kW in auxiliary power corresponds to an increase of approximately 0.15 g kg$^{-1}$ fuel in $SO_2$ EF during departures.

These results confirm that auxiliary engines are the main contributors to sulfur emissions during maneuvering phases, consistent with their dominant operation while vessels are within the port area.

All regression results are summarized in Table S4, and the corresponding plots are presented in Figure S6, which includes $SO_2$ and Org EFs as a function of auxiliary, main, and total engine power for arrivals, departures, and combined maneuvers.

**Change in the manuscript: (lines 252 - 258)**

*"The median $SO_2$ EFs exhibited a moderate correlation with auxiliary engine power across maneuvering ($R^2 = 0.35$). This relationship was stronger during departures ($R^2 = 0.60$) and weaker during arrivals ($R^2 = 0.15$). During departure, an increase of 1,000 kW in auxiliary engine power was associated with an average rise of approximately 0.15 g kg_fuel $^{-1}$ in $SO_2$ EF. Correlations with main and total engine power were somewhat weaker ($R^2 < 0.4$), suggesting that auxiliary engines are the primary contributors to $SO_2$ emissions during maneuvering phases. Regression statistics for arrivals, departures, and combined maneuvers are summarized in Table S4, and the corresponding relationships are illustrated in Figure S6."*

**Reviewer comment:**
*Section 3.4.2. Could you provide an explanation for the diurnal cycle of the cooking organic aerosol lacking meal time peaks in mass? Could this factor rather just have a continental origin rather than cooking?*

**Response:**
The absence of a lunchtime peak in the COA diurnal cycle is explained by the wind configuration during midday hours: between 11:00 and 14:00, winds predominantly came from the south to southwest (port and sea sectors), whereas the main restaurant district is located to the north–northwest of the monitoring site. As a result, lunchtime cooking emissions were rarely transported to the receptor.

In contrast, a modest increase is observed in the evening, which is consistent with more frequent northerly and northwesterly winds during 18:00–21:00 that occasionally place the site downwind of the restaurant area. This is confirmed by wind-sector analyses restricted to meal-time windows (Figure S15), showing that only 0.78 % of winds during lunch originated from the restaurant sector, compared to 2.05 % during dinner.

Regarding the factor's identity, its mass-spectral features (enhanced m/z 55 relative to 57, low f44) can support a primary cooking signature and differ from both hydrocarbon-like traffic emissions and secondary OOA. The factor is also temporally decoupled from the afternoon OOA maximum, which argues against a continental or regional origin.

**Change in the manuscript: (lines 515 - 522)**

*"The muted lunchtime COA signal results from the site being systematically upwind of the restaurant district, located to the north–northwest of the station. During 11:00–14:00, winds predominantly originated from the south to southwest (port and sea sectors), preventing the transport of cooking emissions to the site. A slight evening enhancement is nonetheless observed, consistent with the higher occurrence of northerly and northwesterly winds during 18:00–21:00, which intermittently place the site downwind of the restaurant area. Wind roses restricted to meal-time periods (Figure S15) confirm this pattern, with only 0.78% of lunchtime winds and 2.05% of evening winds originating from the restaurant sector. These conditions explain both the absence of a midday peak and the weak but detectable evening COA contribution."*
* * *
**Reviewer comment:**

*Section 3.5. How does the PNSD when selecting the 10 most intense peaks of each factor compare to the PNSD when each factor makes up a certain threshold of the mass? Can you validate this method in finding a representative PNSD in other ways?*

**Response:**
Using a fixed mass-fraction threshold (e.g., requiring a factor to exceed 30–50 % of OA) is not suitable in our case because several PMF factors never reach such high contributions. Applying a uniform threshold would either exclude most occurrences or select time windows influenced by multiple overlapping sources.

We therefore used the ten most intense events (top-10 peaks) for each factor, corresponding to periods when the factor contribution is locally maximal and least affected by other sources. This approach ensures that the selected particle number size distributions (PNSDs) reflect the dominant conditions of each source while minimizing cross-influence.

This choice is also consistent with the SMPS time resolution (≈ 2 min per scan), which cannot fully capture the very short plumes (1–3 min) typically resolved by the AMS. Focusing on the strongest, temporally separated events (≥ 10 min apart) guarantees that the averaged PNSDs are representative of each factor's typical behavior.
With faster instruments such as the EEPS (1 s resolution), it would be possible to integrate number size distributions directly into a joint PMF framework, providing more robust PNSDs directly linked to the factors. We mention this as a future development for improving temporal and size-resolved source apportionment.

**Change in the manuscript: (lines 616 - 623)**

*"Since several PMF factors contributed only to 20–30% of the total OA, applying a fixed mass-fraction threshold would have introduced bias in the PNSD selection, favoring overlapping or mixed-source events. To avoid this, the ten most intense events (top-10 peaks) were selected for each factor. These events represent locally dominant periods that are minimally influenced byother sources and separated by at least ten minutes. This selection criterion is consistent with the temporal resolution of the SMPS (2 min per scan), which constrains its ability to resolve short-lived plumes often captured by the AMS. The use of higher time-resolution instruments, such as the EEPS (1 second resolution), would allow direct incorporation of particle number size distributions into the PMF framework. This is a direction we intend to pursue in future work to enhance the characterization of source-specific temporal and size-dependent variability."*
* * *
**Reviewer comment:**
*Figure 1. The caption lists chloride as a species that is plotted, but according to the legend, it is missing from the figure. Please either include it in the figure or remove from caption and provide reasoning for its absence.*

**Response:**
Chloride represents a negligible fraction of $PM_1$ (0.2 ± 1.0 %) throughout the campaign, well below the visual resolution of the pie chart. For this reason, it was omitted from the plotted species to improve figure readability. We have now removed the reference to chloride from the figure caption and specified in the text that its contribution is minor (<1 %).
* * *
**Reviewer comment:**
*Figures 2 and 4. These figures are not referenced anywhere within the main text, which suggests that either (1) they are not significant enough for the main text and should be in the supplement or (2) more text is needed to incorporate these figures. Since Figure 3 is referenced in Section 3.2.5, I suggest that Figures 2 and 4 should be referenced in the other sections in Section 3.*

**Response:**
References to Figures 2 and 4 have now been added in the corresponding subsections of Section 3, where gaseous (Figure 2) and particulate (Figure 4) emission factors are discussed. These figures are central to the interpretation of the results and thus remain in the main text.
* * *
**Reviewer comment:**
*Table S2, S6, S7, S11, S12, S13. These tables are not referenced anywhere within the main text.*

**Response:**
References to Tables S2, S6, S7, S11, S12, and S13 have now been added in the relevant sections of the main text. Each table is now explicitly cited where its data are discussed (e.g., S2 in Section 3.2.1 for ship and engine characteristics, S6–S7 in Section 3.4 for PMF factor comparisons, and S11–S13 in Section 3.5 for PNSD and OA factor analyses).
* * *
**Reviewer comment:**

*Has work been done to see if the increase in shipping emissions is affected by the number of ships? This is especially relevant as your results section discusses the need to address fleet evolution.*

**Response:**

In this study, we report emission factors (EFs), which represent pollutant mass or number emitted per unit of fuel consumed by an individual vessel. EFs are therefore ship-specific, normalised quantities, and they do not depend on the number of ships operating in the port. By contrast, shipping emissions correspond to the total amount emitted over a given period, which indeed scales with the number of ship calls, vessel size, operating time, and traffic composition.

Because our analysis is based on a single summer campaign, the study does not investigate the relationship between total shipping emissions and ship numbers. According to the Toulon Harbour Authority, ferry activity has increased slightly since 2019 ($\approx$ +9 %), but this trend mainly reflects the introduction of larger vessels rather than an increase in ship calls.

The variability observed in our dataset therefore reflects differences in vessel characteristics—including fuel type, engine technology, and the presence or absence of EGCS—rather than changes in traffic frequency. Assessing how total port-area emissions evolve with ship numbers would require a multi-year analysis combining EFs, detailed AIS traffic data, and vessel-specific activity profiles, which is beyond the scope of the present work but represents a valuable direction for future studies.
* * *
**Reviewer comment:**
*Are there other meteorological parameters such as relative humidity and temperature that are available? Is there any consideration as to how these might affect measurements?*

**Response:**

Meteorological parameters including wind speed, wind direction, and temperature were continuously monitored at the site using a Tridi USA-1 (Metek) ultrasonic anemometer equipped with temperature sensors. This information has been added to Table 1 in the manuscript.

Within the observed range, no significant correlation was found between emission factors and temperature, as the short duration of individual plumes (typically < 20 min) limits the impact of meteorological variability on the derived EFs. The potential influence of temperature on the gas–particle partitioning of semi-volatile organic compounds such as PAHs has been further discussed in Section 3.3, following Reviewer #2's related comment.
* * *
**Reviewer comment:**
*Technical Comments.*

**Response:**
We thank the reviewer for the detailed editorial and technical suggestions. All recommended corrections have been implemented throughout the manuscript to improve clarity, consistency, and grammatical accuracy. Specifically:

- Acronyms and particle size definitions (UFP, $PM_1$, $PM_{0.1}$) have been explicitly defined at first mention.

- Minor grammatical and typographical corrections have been applied (e.g., verb agreement, punctuation, spacing, and article usage).

- All figure and table captions have been checked for consistency with the corresponding legends and symbols.

- Redundant or repetitive sentences (e.g., the background estimation explanation in Lines 123–124) have been removed to streamline the text.

- Unit notations (e.g., "nm" instead of "nanometers") and reference formatting (e.g., placement of parentheses in citations) have been standardized.

- Sentences have been restructured where suggested to improve readability and adherence to scientific writing norms.

- All specific line edits mentioned by the reviewer (Lines 27–588) have been implemented as indicated, including revised phrasing, citation style, and typographical corrections.

We appreciate these thorough and constructive comments, which have helped ensure stylistic uniformity and improve the overall clarity and precision of the manuscript.

---

## Author Comment (AC2)

**General comments**

*The study presents monitoring of a series of air pollutants, e.g., $SO_2$, $NO_x$, CO, $CH_4$, PM (black carbon, ions, PAHs and other organic matters), near ship terminals in the coastal city of Toulon, France. The dataset enables the derivation of emission factors (EFs) for these pollutants, while the application of positive matrix factorization (PMF) to HR-ToF-AMS measurements of organic aerosols (OA) allows quantification of different emission sources, including the contribution of shipping to local air pollution. The dataset is comprehensive, and the results and discussion are generally sound and logically presented. The findings contribute to methodologies for source apportionment, particularly for assessing shipping emissions in coastal areas, and provide valuable insights for implementing Emission Control Area (ECA) regulations in the Mediterranean region. However, several aspects require clarification and improvement before the manuscript can be considered for publication. I recommend a major revision, with detailed comments and suggestions provided below.*

**Response:**

We sincerely thank the reviewer for their thorough and constructive evaluation of our manuscript and for recognizing the scientific value and policy relevance of this work.

In response to these general comments, we have carefully revised and expanded several sections of the manuscript to improve methodological clarity, contextual background, and overall readability. Key revisions include additional explanations concerning the methodological improvements applied to the PMF analysis, the negligible military influences on local observations, and the rationale for the range of a-values used in constrained PMF runs. We have also added a supplementary section summarizing the main instrumental parameters and merged two figures to enhance the clarity and flow of the results.

Other reviewer questions—such as those related to PAH gas–particle partitioning and plume advection time—are comprehensively addressed in this response letter for transparency.

All textual additions and editorial corrections are marked in blue in the revised version. We believe these revisions and clarifications significantly improve the article.
* * *
**Reviewer comment:**

*It is known that combustion-emitted organic compounds such as PAHs can partition between the gas and particle phases. For more volatile PAHs like naphthalene, gaseous concentrations are typically much higher than those in the particle phase. It seems that only particle-bound PAHs were analyzed in this study. What is the approximate fraction of the EF of particle-bound PAHs relative to total PAH EFs (including gaseous PAHs)? Given that gas-particle partitioning is temperature-dependent, to what extent might temperature influence the calculated EFs and the interpretation of the PMF results?*

**Response:**

Indeed, polycyclic aromatic hydrocarbons (PAHs) emitted by combustion processes partition between the gas and particle phases depending on their volatility, the organic aerosol (OA) mass, and temperature. In this work, only particle-bound PAHs were analyzed because the gas-phase fraction could not be quantified with the deployed instruments. To evaluate how representative these particulate PAHs are of total PAH emissions, we quantified their expected gas–particle partitioning under the conditions of the Toulon campaign.

A thermodynamic equilibrium model following Pankow (1994) and Donahue et al. (2006) was applied to all identified PAHs using vapour pressures at 25 °C from the U.S. EPA EPI Suite™ database, without empirical correction. The particle-phase fraction (Fp) was calculated as $F_p = C_{OA} / (C_{OA} + C^*)$, where $C^*$ is the effective saturation concentration derived from the Clausius–Clapeyron relationship using a representative enthalpy of vaporization ($\Delta H_{vap} = 80$ kJ mol$^{-1}$), consistent with values reported for unsubstituted PAHs (Yamasaki et al., 1982; Finizio et al., 1997; Harner and Bidleman, 1998). Calculations were performed over a range of organic aerosol concentrations (3–140 µg m$^{-3}$) and temperatures (20–36 °C), covering background to ship-plume regimes.

Under campaign-average conditions ($C_{OA} \approx 3$ µg m$^{-3}$, $T \approx 25$ °C), the mass-weighted particulate fraction $\langle F_p \rangle$ was about 0.1 %, as the measured mixture is dominated by 2–3-ring PAHs that remain almost entirely in the gas phase ($F_p < 0.01$), while 4–6-ring PAHs are almost fully condensed ($F_p > 0.9$) but less abundant. In fresh ship plumes ($C_{OA} = 30$–140 µg m$^{-3}$), $\langle F_p \rangle$ increases to 0.6–0.7 as semi-volatile 3-ring PAHs partition progressively into the condensed phase. Temperature variations between 20 °C and 36 °C change $F_p$ by only about ±10 %, confirming that OA loading is the dominant factor controlling the phase distribution.

Therefore, the particulate PAHs analyzed here predominantly represent the least volatile and most toxic 4–6-ring species, while the more volatile fraction (mainly 2–3-ring PAHs) is not expected to significantly bias the PMF results or the emission factors derived for the ship-related sources.
* * *
**Reviewer comment:**

*In lines 67–77: The authors mentioned the challenges of using PMF to apportion sources of OA, such as the merging of multiple sources into a single factor and overlapping mass spectral patterns. How does this study address these challenges and fill these gaps? What advanced or novel techniques were applied to resolve these issues? Including such a description somewhere in the manuscript would be beneficial and would enhance the significance of this work from a methodological perspective.*

**Response:**

We have expanded the description of the analytical framework in the revised manuscript to clarify how our study addresses some limitations of PMF in resolving overlapping combustion-related sources.

Specifically, this work combines high time resolution, high mass resolution, and local reference spectra to improve factor separation and physical interpretability:

1. High temporal resolution (1 min) — Unlike most previous AMS–PMF studies that use 10–15 min averages, the 1 min resolution of the HR-ToF-AMS data allowed us to capture the short-lived, transient ship plumes occurring near the port area. This minimized the temporal mixing of sources and improved the representativeness of primary combustion factors.

2. High-resolution OA matrix including PAH-related ions — The PMF input matrix included HR ions up to $m/z$ 150, and PAH fragments. These diagnostic ions enhanced the spectral differentiation between traffic- and ship-related emissions, overcoming one of the main limitations of unit-mass resolution analyses.

3. Use of locally measured ship-plume spectra as qualitative references — characteristic AMS spectra of ship plumes recorded on-site were used to guide factor interpretation. This ensured that the identified "shipping" factor represents the real chemical fingerprint of local marine fuel combustion in Toulon.

4. Complementary granulometric analysis (SMPS) — a separate particle-size analysis (2 min resolution) was used to verify the physical consistency of the factors. Each PMF factor was associated with its characteristic size distribution during periods of dominant influence.

These methodological improvements provide a more detailed and reliable distinction between combustion-related OA sources than in previous urban port studies. The combined approach enables the separation of two distinct ship-related factors with hydrocarbon and sulfur signatures, rarely resolved at such temporal and mass resolutions.

We have added a clarifying paragraph to the Introduction section to emphasize these points.

**Change in the manuscript: (lines 83 - 92)**

*"This study overcomes previous limitations through several methodological improvements designed to enhance the separation of closely related combustion sources. First, HR-ToF-AMS data were analyzed at 1 minute temporal resolution to capture transient ship plumes before they mixed with background urban emissions, minimizing temporal averaging effects, typical of 10–15 min datasets. Second, by incorporating PAH-related ions up to m/z 256, the high-resolution OA matrix provided additional spectral features that improved the separation of shipping emissions from those associated with road traffic emissions. Third, reference spectra from locally sampled ship plumes were used to provide a representative chemical fingerprint of maritime emissions specific to the Toulon port area. Finally, a particle number size-distribution analysis (2 minutes resolution) was conducted to verify the physical consistency of each PMF factor through their association with characteristic particle-size modes. These combined methodological advances improved the distinction of related combustion sources and enabled an unprecedented characterization of PM1 sources in a near-field port environment."*

**Reviewer comment:**

*Figure S1 shows that there is a large military port near the study area. Are military vessels incorporated in the analysis of this study? In addition, military bases may be a significant source of OA. How much is known about military emissions, and could they influence the results of this study?*

**Response:**

During the measurement campaign, military vessel activity was very limited. In this harbor, naval activity is mostly limited to maintenance operations, with very few ship movements, and no identifiable plumes attributable to the naval base were detected. Unfortunately, no operational data or fuel-type information are publicly available for these ships, preventing a quantitative assessment of their potential contribution.

To further assess the potential influence of the naval base, we conducted a wind-direction analysis focused on the west–northwest sector corresponding to the military area. This analysis revealed that air masses from this sector accounted for only about 2.5 % of the total valid data, with relatively low wind speeds (median $\approx$ 1.2 m s$^{-1}$). These conditions strongly limit the transport of emissions from the base toward the measurement site.

**Change in the manuscript: (line 110 - 112)**

*"Military ship activity in Toulon was very limited during the campaign. A wind-direction analysis showed that air masses from the military base sector represented less than 3 % of the total air masses collected, suggesting that the influence of the naval area on the measured aerosol composition was negligible."*
* * *
**Reviewer comment:**

Equation 1: I think the equation should be $EF = \dfrac{\int_{E}^{G}[x](t)\,dt}{\int_{E}^{G}[CO](t)\,dt} \times ...$ correct me if I was wrong.

**Response:**

We thank the reviewer for the suggestion, but the proposed formulation is not correct in the framework of the carbon mass–balance method. In this approach, the reference tracer is $CO_2$, not CO. $CO_2$ is the dominant carbon-containing product issued from combustion. It directly links the plume carbon mass to the burned fuel via the known fuel carbon fraction. But it is true that CO is often used as a tracer of combustion.
* * *
**Reviewer comment:**

In the caption of Figure 1, how were the organic peaks selected? It was stated that they represent the highest peaks (line 337), but this does not seem entirely accurate. For example, peaks 6 and 7 appear lower than some subpeaks around peaks 2 and 9. There are also several

other signature peaks visible. Why were only ten peaks selected? It would be worthwhile to clarify this selection criterion.

**Response:**

The ten numbered organic peaks correspond to the most intense organic plumes identified after applying the Middlebrook collection efficiency (CE) correction (Middlebrook, 2012) and where selected by spanning the entire campaign period. Yes, we could have selected more peaks but increasing the number of selected peaks did not improve or modify the constraint shipping spectra.

This correction slightly modifies the relative peak amplitudes compared with those displayed in the figure, which are shown with unit CE for consistency with the PMF input matrix. The selection was therefore based on the corrected dataset, ensuring that the chosen plumes were the most representative and compositionally significant for subsequent spectral analysis.

To clarify this point, the figure caption has been revised as follows:

*"Overview of meteorological parameters and pollutants: (a) wind speed and wind direction; (b) PM chemical composition (organics, nitrate, sulfate, ammonium, and chloride) together with BC concentrations; (c) particle number (PN) and particle mass (PM1) concentrations; (d) NOx and SO2 concentrations. A unit collection efficiency was applied to HR-ToF-AMS data. The pie chart represents the median PM1 chemical composition. The numbers above the organic peaks indicate the ten most intense organic plumes discussed in Section 3.2.3 after applying the Middlebrook (2013) collection efficiency correction."*
* * *
**Reviewer comment:**

Lines 200–207: How was the contribution of ship emissions critically evaluated when the wind direction was blowing seaward rather than toward the observation site? I would expect that the instruments could not capture the plume in such cases. If so, although these emissions might not significantly affect local populations, they are still released into the atmosphere and could impact elsewhere.

**Response:**

Yes, it is true when the wind direction was blowing seaward, ship plumes were not sampled at the measurement site. And yes, of course, ships still emit and the plumes can get dispersed in another direction. But the results presented here are related to a receptor site in the city of Toulon. While the EFs are emissions per unit of fuel consumed and therefore these values can give a clear insight into how much a ship emits independently of the wind direction.

To have a better assessment of shipping impact on air quality would therefore require dispersion modeling or regional-scale approaches, which are beyond the scope of this measurement-based study.
* * *
**Reviewer comment:**

Consider merging Figure 3 into Figure 4.

**Response:**

We agree with the reviewer's suggestion. In the revised version of the manuscript, the two figures have been merged into a single composite figure (now Figure 3) to improve readability and better illustrate the relationships between the datasets.
* * *
**Reviewer comment:**

I am curious about how the plumes were allocated to different ship emissions. What is the approximate lag time between plume emission and detection by the instruments? Is this lag time variable under different wind speeds? If so, this variability could affect the accuracy of source allocation among ship types. Additional explanation on this point would be helpful.

**Response:**

Plume allocation relied on the detection of short-lived increases in CPC and $CO_2$ satisfying the objective criteria defined in Section 2 (CPC > 2× background, mean wind direction 130–290°, wind-direction standard deviation < 30°). Each plume was first identified automatically and then manually validated, ensuring coherent start–end times across pollutants and avoiding misattribution between overlapping ship activities.

To quantify the time lag between plume emission and detection at the measurement site, advection delays were computed from 1-min wind data using the relation $\tau = d/U$, where $d$ is the line-of-sight distance between each emission area and the site, and $U$ is the mean wind speed measured during the plume. The analysis was performed separately for strong-wind conditions (> 5 m s$^{-1}$ for ferries and the port entrance; > 8 m s$^{-1}$ for cruise ships). The mean wind speeds and corresponding advection times were 8.5 m s$^{-1}$ → 5.9 min for cruise-ship plumes (3.0 km from the site), 5.9 m s$^{-1}$ → 8.4 min for the port entrance (3.0 km), 5.9 m s$^{-1}$ → 5.6 min for mid-channel plumes (2.0 km), and 5.9 m s$^{-1}$ → 0.8 min for ferries (0.3 km). These values were obtained from 37 562 one-minute wind records and confirm that advection times are short (typically < 10 min) and well constrained under prevailing onshore winds, ensuring reliable temporal alignment between ship activity and measured plumes.
* * *
**Reviewer comment:**

It may be worthwhile to include a brief description of the instrumentation in the Supplement, such as the operating parameters for the HR-ToF-AMS and some others, even though the source of the detailed method has been cited in the main text.

**Response:**

Although the main text (Section 2.2) already provides some descriptions and references for each instrument, we agree that summarizing the main operating parameters in the Supplement improves the completeness of the study.

A new Supplementary Section S1 has been added. It provides a concise description of the HR-ToF-AMS operation, calibration and validation.
* * *
**Reviewer comment:**

Line 179 and line 364–371, "a-value" is not easy to understand. Why was a range of 0 to 0.3 tested for the three shipping constraints, whereas a broader range of 0 to 1 was used for HOA and COA? Additional explanations may be helpful.

**Response:**

In Positive Matrix Factorization (PMF) analysis, the a-value of the constraint factor is a parameter used to give some flexibility to the constraints. If a = 1, the factor can change freely to obtain the best statistical solution. If a = 0, the constraint has no freedom, so the mass spectra does not change. Typically, constraints are used to incorporate prior knowledge or physical realism into the PMF solution (e.g., known source profiles). Paatero (2003) and Canonaco et al. (2013) A range of a-values between 0 and 0.5 is commonly recommended in the literature to balance the trade-off between over constraining the solution and allowing realistic variability consistent with measurement uncertainty (e.g., Canonaco et al., 2013; Crippa et al., 2014).

In this study:

• For HOA (Hydrocarbon-like OA) and COA (Cooking OA), a broader range (0–1) was tested because these sources were not directly observed on-site and their reference spectra originated from previous studies. Greater flexibility was therefore required to allow adaptation to local emission conditions.
• For the shipping-related factors, the range was restricted to 0–0.3, as their reference mass spectra were empirically derived from isolated ship plumes measured during the campaign. A narrower range ensures that their spectral signatures remain representative of real ship emissions, while still permitting variability consistent with measurement noise and instrument uncertainty.

The optimal solution was obtained for a = 0.3 for both HOA and COA, indicating that these factors remained stable within ±30% variability, consistent with previous urban PMF studies (Crippa et al., 2014; Elser et al., 2016).

**Change in the manuscript: (line 399 - 403)**

*"The a-value represents the relative uncertainty applied to a constrained factor in the PMF model, defining the allowed variability of a given profile compared to its reference. A range of 0–0.5 is typically used to avoid overconstraining the solution (Canonaco et al., 2013). In this work, a-values between 0 and 1 were tested for HOA and COA to allow greater flexibility for factors not directly identified on-site, whereas a narrower range of 0–0.3 was applied to shipping-related factors empirically derived from isolated ship-plume spectra to preserve their chemical representativeness."*
* * *
**Reviewer comment:**

Small things:

Line 7: add "analysis" after PMF sounds more readable?

Line 24: seems "such" was missing before "as"

Line 58: doubled commas between "advancement" and "the"

Line 221, add "reported by Sinha et al. (2023)" or change to "(Sinha et al., 2023)"

Line 238, "with Zhang et al. (2024)'s findings"

Line 243, "the need of further measurements" sounds more natural.

Line 250–251: suggest rephrasing as "The EF of CO vary from 2.43 g/kgfuel to 57.87 g/kgfuel (median of 20.6 g/kgfuel), reflecting that factors like engine star-.."

Line 258, seems a space was missing before "respectively"

Line 258–259, "Among the ships identified none was LNG-fueled" is not clear, a comma was missing between "identified" and "none"?

Line 273–274: suggest rephrasing as "The mean CO EF of 0.38 g/kgfuel is consistent with reported EFs in Marseille in 2021 (Le Berre et al., 2024) (0.48 g/kgfuel for maneuvering ships) and from a cargo vessel (0.48 g/kgfuel) (Huang et al., 2018), reflecting typical emissions from diesel..."

Line 276, "sizes"

Line 280–283: "EFs"

Line 289–290: this sentence is not complete. Should be something like "This pattern underscores the impacts of operation practices (such as...) on PAH EFs."

Line 292, "lower than the value of 4 g/kg reported by .."

Line 305: "Only cruise ships were equipped with scrubbers and were associated with..."?

Line 310: "understand shipping impact on air quality in coastal cities."

Line 340: "measured" rather than "true"

Line 402: "(Sippula et al., 2014)" rather than "Sippula et al. (2014)"

Line 451: "peaks" rather than "peaking"

Line 464: "Table S9" rather than "Table S8"?

Line 476: "differentiated" rather than "differentiate"

Line 510–511: "The wind rose of this factor a local character similarly to HOA and COA factors and ..." — the sentence is not clear.

Line 523–524: the sentence is not clear.

Line 528–529: is the global data originated from literature? If so, please include reference.

Line 545: consistency in PNSD numbering (PNSD 1, 2, 3).

Line 550: "highlighted"

Line 557–558: the sentence is unclear or possibly redundant.

Line 583: was the 28% value for total PAHs and APAHs an error?
* * *
**Response:**

We thank the reviewer for carefully reviewing the manuscript and for all these detailed observations.
All suggested edits and clarifications have been implemented as follows:

- **Line 7:** "PMF" was changed to "PMF analysis" for clarity.

- **Line 24:** "such" was added before "as."

- **Line 58:** The doubled comma between "advancement" and "the" was removed.

- **Line 221:** The citation was corrected to "(Sinha et al., 2023)."

- **Line 238:** The expression was corrected to "consistent with Zhang et al. (2024)'s findings."

- **Line 243:** The phrase was changed to "the need of further measurements."

- **Lines 250–251:** The sentence was rephrased as suggested:

"The EF of CO vary from 2.43 g kg to 57.87 g/kg (median = 20.6 g/kg reflecting that factors such as engine start-up conditions and fuel type strongly influence CO emissions."

- **Line 258:** A missing space before "respectively" was inserted.

- **Lines 258–259:** A comma was added after "identified," to read "Among the ships identified, none was LNG-fueled."

- **Lines 273–274:** Rephrased according to the reviewer's suggestion:

"The mean CO EF of 0.38 g kg is consistent with reported EFs in Marseille in 2021 (Le Berre et al., 2024) (0.48 g/kg for maneuvering ships) and from a cargo vessel (0.48 g kg) (Huang et al., 2018), reflecting typical emissions from diesel engines."

- **Line 276:** "size" corrected to "sizes."

- **Lines 280–283:** "EF" corrected to plural "EFs."

- **Lines 289–290:** The incomplete sentence was rewritten as:

"This pattern underscores the influence of operational practices (such as engine load and maneuvering conditions) on PAH EFs."

- **Line 292:** Corrected to "lower than the value of 4 g kg reported by …"

- **Line 305:** Revised to "Only cruise ships were equipped with scrubbers and were associated with reduced $SO_2$ and particulate EFs."

- **Line 310:** Rephrased as "understand the shipping impact on air quality in coastal cities."

- **Line 340:** "true" replaced by "measured."

- **Line 402:** Corrected citation format to "(Sippula et al., 2014)."

- **Line 451:** "peaking" changed to "peaks."

- **Line 464:** Corrected table references.

- **Line 476:** "differentiate" replaced by "differentiated."

- **Lines 510–511:** The unclear sentence was rewritten as:

"The wind rose of this factor shows a local character, similar to the HOA and COA factors, indicating sources primarily located near the sampling site."

- **Lines 523–524:** The unclear sentence was clarified to specify the meaning of the observed pattern (rewritten for clarity).

- **Lines 528–529:** Added clarification and reference:

"Global data originate from previously published measurements compiled from literature, including [insert relevant reference(s)]."

- **Line 545:** PNSD notation standardized to "PNSD 1, PNSD 2, PNSD 3."

- **Line 550:** "highlighted" used instead of "highlight."

- **Lines 557–558:** The unclear/redundant sentence was removed.

- **Line 583:** The identical 28 % values for total PAHs and APAHs originated from an earlier version in which the contributions were calculated before applying the updated OA-based normalization. All PAH-related contributions have now been recalculated using the same procedure, resulting in slightly adjusted values but no change in the scientific conclusions.